# Current CRISPR gene drive systems are likely to be highly invasive in wild populations

Charleston Noble[1,2,3†], Ben Adlam[1,4†], George M Church[2,3], Kevin M Esvelt[5]*, Martin A Nowak[1,6,7]*

[1]Program for Evolutionary Dynamics, Harvard University, Cambridge, United States; [2]Department of Genetics, Harvard Medical School, Harvard University, Boston, United States; [3]Wyss Institute for Biologically Inspired Engineering, Harvard University, Boston MA, United States; [4]School of Engineering and Applied Science, Harvard University, Cambridge, United States; [5]Massachusetts Institute of Technology Media Lab, Cambridge, United States; [6]Department of Mathematics, Harvard University, Cambridge, United States; [7]Department of Organismic and Evolutionary Biology, Harvard University, Cambridge, United States

*For correspondence:
esvelt@mit.edu (KME);
martin_nowak@harvard.edu
(MAN)

†These authors contributed
equally to this work

Competing interests: The
authors declare that no
competing interests exist.

Reviewing editor: Michael
Doebeli, University of British
Columbia, Canada

**Abstract** Recent reports have suggested that self-propagating CRISPR-based gene drive systems are unlikely to efficiently invade wild populations due to drive-resistant alleles that prevent cutting. Here we develop mathematical models based on existing empirical data to explicitly test this assumption for population alteration drives. Our models show that although resistance prevents spread to fixation in large populations, even the least effective drive systems reported to date are likely to be highly invasive. Releasing a small number of organisms will often cause invasion of the local population, followed by invasion of additional populations connected by very low rates of gene flow. Hence, initiating contained field trials as tentatively endorsed by the National Academies report on gene drive could potentially result in unintended spread to additional populations. Our mathematical results suggest that self-propagating gene drive is best suited to applications such as malaria prevention that seek to affect all wild populations of the target species.
DOI: https://doi.org/10.7554/eLife.33423.001

## Introduction

CRISPR-based gene drive systems can bias inheritance of desired traits by cutting a wild-type allele and copying the drive system in its place (*Esvelt et al., 2014*). Following reports of successful CRISPR gene drive systems in yeast (*DiCarlo et al., 2015*) and fruit flies (*Gantz and Bier, 2015*), scientists emphasized the need to employ strategies beyond traditional barrier containment as a laboratory safeguard (*National Academies of Sciences, Engineering, and Medicine, 2016*; *Akbari et al., 2015*). These precautions were judged necessary to prevent unintended ecological effects, but also because any unauthorized release affecting a wild population could severely damage trust in scientists and governance, significantly delaying or even precluding applications of gene drive and other biotechnologies.

Drive resistance can result from mutations that block cutting by the CRISPR nuclease. Recent examinations of the phenomenon by experiments and deterministic models have generated substantial media attention (*Champer et al., 2017*; *Unckless et al., 2017*; *Drury et al., 2017*; *Noble et al., 2017*). Resistance can arise from standing genetic variation at the drive locus or because the drive mechanism is not perfectly efficient and is predicted to prevent drive fixation in wild populations

**eLife digest** Gene drive is a genetic engineering technology that can spread a particular suite of genes throughout a population. Among the types of gene drive systems, those based on the CRISPR genome editing technology are predicted to be able to spread genes particularly rapidly. This is because components of the CRISPR system can be tailored to replace alternative copies of a particular gene, ensuring that only the desired version is passed on to offspring. In this way, for example, a gene that prevents mosquitoes from carrying or transmitting the malaria parasite could be introduced to a very large wild population to reduce the incidence of the disease among humans.

Gene drives can be "self-propagating" or "self-exhausting": the former are designed so that they can always spread as long as there are wild organisms around, whereas the latter are expected to lose their ability to spread over time. Self-propagating CRISPR gene drives have been shown to work in controlled populations of fruit flies, mosquitoes and yeast. These experiments happen in a controlled environment in the laboratory, so the organisms edited to have the gene drive elements do not come in contact with susceptible wild organisms. However, if just a few were to escape, the gene drive could theoretically spread quickly outside the laboratory.

Noble, Adlam et al. investigated, using mathematical models, whether or not – and how fast – a self-propagating CRISPR-based gene drive would spread if a number of organisms with the gene-drive elements were released into the wild. The models showed that the release of just a few of the edited organisms would result in the gene drive spreading to most populations that interbreed. This happened regardless of the structure of the wild populations or whether a degree of resistance to the drive emerged. As a result, even the smallest breach of a contained trial could lead to significant gene drive spread in the wild.

The findings suggest that self-propagating gene drive technologies would be most useful where the invasion of most wild populations of the target species is the intended purpose, rather than a risk to be avoided. As a result, a self-propagating CRISPR-based gene drive could be well suited to spreading among mosquitoes to impede the malaria parasite, provided there were strong international agreements in place. The findings also underline the difficulty of carrying out safe field trials of self-propagating gene drives, and the need for very tight control of laboratories carrying out experiments in this field. Lastly, they highlight the importance of developing and testing the evolutionary stability of self-exhausting gene drives, which could be better contained to local populations.

DOI: https://doi.org/10.7554/eLife.33423.002

unless additional mitigating strategies are employed (*Burt, 2003*; *Deredec et al., 2008*; *Esvelt et al., 2014*; *Noble et al., 2017*; *Marshall et al., 2017*). Recent articles highlighting the problem of resistance for self-propagating gene drives have suggested that it might prevent drive invasion in wild populations—with some even implying that resistance could serve as an experimental safeguard. While we agree that resistance should prevent drive fixation, an allele can nonetheless spread to significant frequency without fixing. To clarify this point, we sought to quantify the likelihood and magnitude of spread in the most likely unauthorized release scenario—a small number of engineered individuals released into a wild population.

CRISPR-based gene drive systems function by converting drive-heterozygotes into homozygotes in the late germline or early embryo (*Esvelt et al., 2014*) (*Figure 1A*). First, a CRISPR nuclease encoded in the drive construct cuts at the corresponding wild-type allele—its target prescribed by an independently expressed guide RNA (gRNA)—producing a double-strand break (*Jinek et al., 2012*). This break is then repaired either through homology-directed repair, producing a second copy of the gene drive construct, or through a nonhomologous repair pathway (non-homologous end joining, NHEJ, or microhomology-mediated end joining, MMEJ), which typically introduces a mutation at the target site (*Mali et al., 2013*; *Cong et al., 2013*). Because the drive target is determined through sequence homology, such a mutation generally results in resistance to future cutting by the gene drive. Thus, the allele converts from a wild-type to resistant allele if it undergoes repair by a pathway other than homology-directed repair. Moreover, drive-resistant alleles are expected to

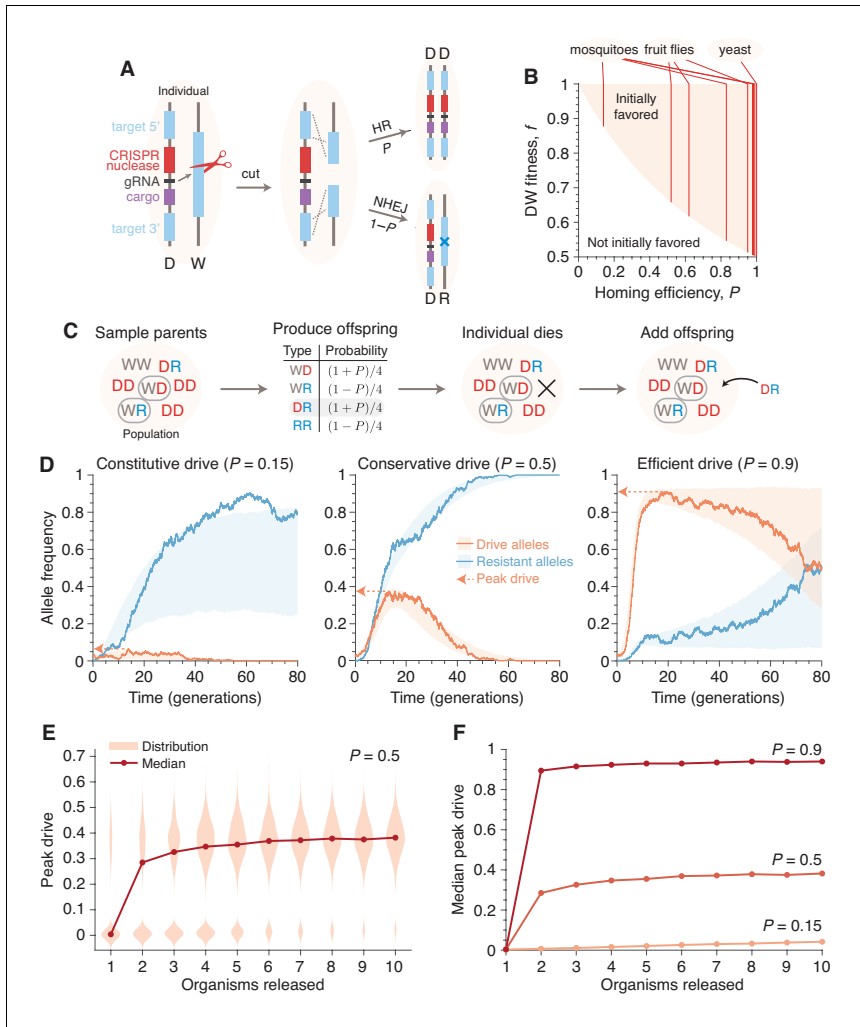

**Figure 1.** Existing alteration-type CRISPR gene drive systems should invade well-mixed wild populations. (A) Typical construction and function of alteration-type CRISPR gene drive systems. A drive construct (D), including a CRISPR nuclease, guide RNA (gRNA), and 'cargo' sequence, induces cutting at a wild-type allele (W) with homology to sequences flanking the drive construct. Repair by homologous recombination (HR) results in conversion of the wild-type to a drive allele, or repair by nonhomologous end-joining (NHEJ) produces a drive-resistant allele (R). (B) Drives are predicted to invade by deterministic models when the fitness of DW heterozygotes, $f$, and the homing efficiency, $P$, are in the shaded region. Vertical lines indicate empirical efficiencies from Appendix 1—table 1. (C) Diagram of a single step of the gene-drive Moran process. (D) Finite-population simulations of 15 drive individuals released into a wild population of size 500, assuming conservative ($P = 0.5$) or high ($P = 0.9$) homing efficiencies, as well as a low-efficiency, constitutively active system ($P = 0.15$). Individual sample simulations (solid lines), and 50% confidence intervals (shaded), calculated from $10^3$ simulations. Drive-allele frequencies red and resistant-allele frequencies blue. Peak drive, or maximum frequency reached, is illustrated by dashed lines and arrows. (E) Peak drive distributions and medians with varying numbers of individual organisms released ($P = 0.5$). (F) Medians of peak drive distributions for varying homing efficiencies ($P = 0.15$, bottom; $P = 0.5$, middle; $P = 0.9$, top). Throughout, we assume neutral resistance ($f_{WR} = f_{RR} = 1$) and a 10% dominant drive fitness cost ($f_{WD} = f_{DD} = f_{DR} = 0.9$).

DOI: https://doi.org/10.7554/eLife.33423.003

exist in wild populations simply due to standing genetic variation (*Unckless et al., 2017*; *Drury et al., 2017*).

Deterministic models, which assume an infinite, well-mixed population, predict whether an allele is predicted to increase in frequency when initially rare in a wild population. Whether gene drives are predicted to invade by deterministic models depends on two key parameters: the homing efficiency

($P$), or the probability of undergoing homology-directed repair instead of nonhomologous repair, and fitness ($f$), or the relative fecundity or death rate the drive and its cargo confer on their organism compared to the wild-type. Mathematically, drives are initially favored by selection if $f(1 + P) > 1$, i.e., if the inheritance bias of the drive exceeds its fitness penalty (*Noble et al., 2017*; *Deredec et al., 2008*; *Unckless et al., 2015*). Given that the homing efficiencies of reported drive systems typically range from 0.37 to 0.99 (*Appendix 1—table 1*), current drive systems can clearly invade in deterministic models. Although the fitness parameter, $f$, is typically not measured in proof-of-concept studies, a substantial fitness cost is tolerable by all reported CRISPR drive constructs (*DiCarlo et al., 2015*; *Gantz and Bier, 2015*; *Champer et al., 2017*; *Gantz et al., 2015*; *Hammond et al., 2016*) (*Figure 1B*).

However, in finite populations, the fate of initially rare alleles is determined not only by selection but also by stochastic fluctuations (*Wright, 1931*; *Fisher, 1930*; *Haldane, 1927*). Therefore, stochastic models are required to predict the probability that a drive will invade a population upon the introduction of a very small number of individuals, even when deterministic models predict that they are to invade. A previous, and arguably prescient, stochastic model of endonuclease drive containment found that homing-based drives, such as those subsequently developed using CRISPR, were among the likeliest to invade of the various drive alternatives (*Marshall, 2009*). To determine whether self-propagating homing drives are still able to invade in the presence of resistance, we formulated a finite population, stochastic, Moran-based model that allows us to study small releases in finite and structured populations (Materials and methods).

## Results

Our model considers three distinct allelic classes: wild-type (W), gene drive (D), and resistant (R). Consistent with experiments, we assume that the drive invariably cuts the wild-type allele in the germline of a heterozygous WD individual, converting to a drive allele with probability $P$, or a resistant allele with probability $1 - P$. Each genotype, AB, has a relative reproductive rate, $f_{AB}$, corresponding to its fitness in deterministic models, normalized such that the wild-type homozygote has fitness one ($f_{WW} = 1$), the drive confers a dominant cost ($f_{DW} = f_{DD} = f_{DR} < 1$), and resistance is neutral ($f_{WR} = f_{RR} = 1$). This ordering of the parameters conservatively represents the worst-case scenario for drive spread (Comparison with deterministic model).

At the population level, our basic model considers $N$ diploid individuals mating randomly. The process unfolds in discrete steps, during which parents are chosen for reproduction, an offspring is chosen according to the mechanism above, and another individual is replaced by the offspring (*Figure 1C* and Materials and methods). These steps are repeated until one allele fixes. A generation is $N$ time-steps, which corresponds to the mean lifespan of an individual.

Code to perform numerical simulations of this model and all model extensions described below (C++, Matlab), as well as data files, documentation, and code to reproduce all of the figures shown here (Matlab) can be found at GitHub (*Noble, 2018;* copy archived at https://github.com/elifescien-ces-publications/drive-invasiveness).

*Figure 1D* shows typical simulations for drive efficiencies of $0.15, 0.5$, and $0.9$, which correspond respectively to a constitutively active drive system targeting a common insertion site, and conservative and high efficiency systems (based on previous experimental studies, *Appendix 1—table 1*, *Figure 1B*, Empirical data supplement). These simulations assume a dominant drive fitness cost of 10%, a population of size 500, and a release of 15 drive-homozygous individuals. (Note that the dynamics are similar for larger population sizes; see Population size and Figure 3). In all three cases, the drive, on average, irreversibly alters a majority of the population, either via invasion of the drive itself or via spread of drive-created resistant alleles. We call the maximum frequency of drive alleles reached during a simulation the peak drive, and we say a drive has invaded if it establishes in the population, ensuring behavior qualitatively similar to deterministic models (Comparison with deterministic model). Notably, for sufficiently large populations, arbitrarily low frequencies meet this standard, as it depends on the absolute number of drive alleles rather than their frequency (Analytic formulae for the escape probability in structured populations). Note also that each of these examples is chosen from the parameter regime in which invasion is predicted by deterministic models, since invasion is very unlikely outside of this regime.

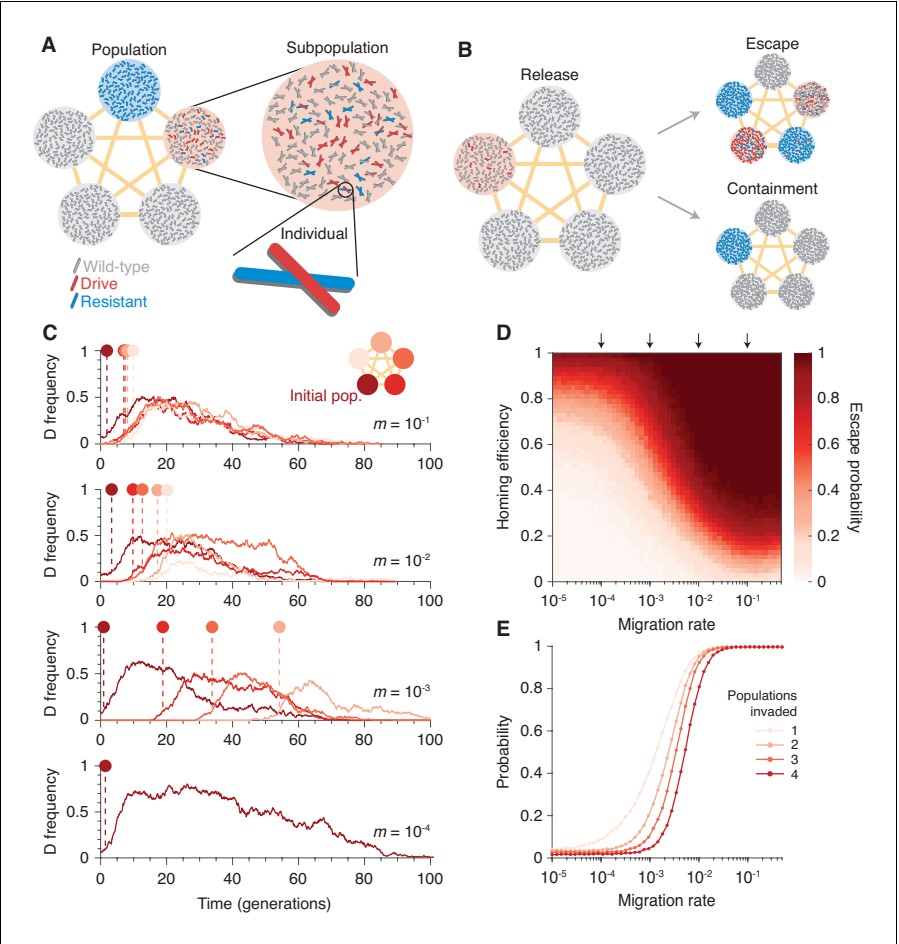

**Figure 2.** Existing CRISPR gene drive systems should invade linked subpopulations connected by gene flow. (**A**) Diagram of well-mixed subpopulations (circles) linked by gene flow (edges). Individuals represented by chromosomes with wild-type (gray), drive (red), or resistant (blue) haplotypes. (**B**) Few drive homozygotes are released in one subpopulation. The drive escapes if it invades another subpopulation before going extinct. Otherwise it is contained. (**C**) Typical simulations for varying migration rates ($m = 10^{-1}$, top, to $m = 10^{-4}$, bottom), following introduction into a single subpopulation. Lines represent drive frequencies in each subpopulation. Circles correspond to the time the drive invades a subpopulation. Population color is by invasion order, not predetermined. (**D**) Escape probability as a function of homing efficiency, $P$, and migration rate, $m$. Arrows indicate migration rates from B. Each pixel is calculated from $10^3$ simulations. (**E**) Probability of invading 1, 2, 3, or 4 additional populations (aside from the originating population, which is typically invaded), assuming a homing efficiency of $P = 0.5$. Each data point is calculated from $10^4$ simulations. Throughout, we consider five subpopulations connected in a complete graph, each consisting of 100 individuals. Initially, 15 drive homozygotes are introduced into one subpopulation. Resistance is neutral ($f_{WR} = f_{RR} = 1$) and the drive confers a dominant 10% cost ($f_{WD} = f_{DD} = f_{DR} = 0.9$).

DOI: https://doi.org/10.7554/eLife.33423.004

We next calculated the distribution of peak drive while varying the number of organisms released (*Figure 1E and F*). We find that these distributions are bimodal, with one mode centered around the initial frequency (corresponding to drift leading rapidly to extinction) and one centered roughly around the maximum values observed in the large-release scenarios in *Figure 1D*. The former mode shrinks rapidly as more organisms are released, and for the parameters studied, a release of 10 individuals nearly guarantees invasion with substantial peak drive (Comparison with deterministic model, Figure 10).

To understand the extent to which isolation might prevent invasion of other populations connected by gene flow, we introduced population structure. Our model consists of five subpopulations

(or islands) that are equally connected by migration (*Figure 2A* and Finite population model with population structure). Typical dynamics are illustrated in *Figure 2C*. *Figure 2B and D* show the escape probability, or the probability of the drive invading (arbitrarily defined as attaining a frequency of 0.1) at least one subpopulation other than its originating one, and *Figure 2E* shows the probability of invading a varying number of subpopulations.

Our results in *Figure 2* suggest that if the migration rate is extremely low, then the drive is effectively contained in the initial subpopulation. If the migration rate is high, the drive is almost guaranteed to invade all subpopulations linked to the originating one. For intermediate migration rates—characterized roughly by migration rates on the order of the inverse of the drive extinction time—both outcomes occur. In the scenario studied in *Figure 2*, a migration rate of $10^{-3}$, which corresponds to a single migration event every 2 generations on average (Materials and methods), virtually guarantees escape for moderate drive efficiencies (Materials and methods). For further details and analytical formulae allowing rapid estimation of escape probabilities, see Analytic formulae for the escape probability in structured populations.

Finally, we sought to understand the effects of additional mitigating factors that could potentially affect peak drive or invasion. We considered the most prominent factors that have arisen in previous papers, and we studied each by varying parameters in our basic model and developing model extensions. Our results are explored in detail in the Materials and methods.

First, we considered preexisting drive resistance resulting from standing genetic variation (*Unckless et al., 2017*; *Drury et al., 2017*) (Standing genetic variation). We find that increasing the proportion of the population that is initially resistant linearly decreases the mean peak drive ($R^2 = 0.996$). Using the parameters in *Figure 1E* and considering a release of 15 individuals, more than 50% preexisting resistance is required to contain average peak drive below 10% (*Figure 4*).

Second, we studied the effect of varying family size, which may be relevant to species such as mosquitoes with large egg batch sizes (*Hammond et al., 2016*; *Yaro et al., 2006*). We extended the model so that $k$ (adult) offspring are produced from a reproduction event, rather than one. We find that this effect scales the release and population sizes (*Hill, 1972*) by a factor of $4/(2k + 6)$. For illustration, we estimated $k$ for Anopheles gambiae to be roughly 10 (Offspring number distribution), so that a release of 7 individuals roughly corresponds to a release of 1 individual in our basic model. While this effect somewhat reduces the chance of drive invasion for small release sizes, it does not preclude it.

Third, we varied drive fitness, resistant-individual fitness and homing efficiency across their entire parameter regimes and recorded peak drive (Effect of varying fitness and homing efficiency, *Figure 7*, *Figure 8*). While varying drive fitness, we find that peak drive is on average greater than 30% across the majority of the regime and almost always greater than 10% (*Figure 7*, left)—and, as a technical aside, we find that this is the case whether the fitness cost of the drive manifests itself via a reduction in birth rate or via increase in death rate (*Figure 7*, right). Moreover, in line with previous deterministic results, we find that peak drive can be substantially increased by associating a fitness cost with resistance (*Figure 8*), which could be expected for drive constructs intended for large-scale application, utilizing methods such as multiplex targeting of essential genes (*Esvelt et al., 2014*; *Noble et al., 2017*; *Marshall et al., 2017*).

Fourth, we studied the effect of inbreeding, which has been shown in several recent theoretical studies (*Bull, 2017*; *Drury et al., 2017*) to impede drive spread (Inbreeding). We extended the model to include a probability $s$ of an individual selfing rather than mating with a second individual (*Bull, 2017*). The model assumes no inbreeding depression and thus considers the worst-case scenario for drive (*Bull, 2017*). We find that even in this scenario, high selfing probabilities are required to reduce peak drive and the probability of invasion for moderate drive costs.

There are a variety of other phenomena that could affect invasiveness, e.g., density dependence (*Deredec et al., 2011*), environment (*Tanaka et al., 2017*), costly resistance (*Traulsen and Reed, 2012*), local ecology, and even mating incompatibilities between some laboratory strains and wild individuals. Such effects should be carefully studied in subsequent papers. Most importantly, the drive architecture itself should affect invasiveness; we consider here only alteration-type drive systems, while others, e.g., sex-ratio distorters and genetic load drives, would be expected to yield different dynamics. In particular, population suppression drive systems may locally self-extinguish before invading new populations. However, for alteration drives, our key qualitative finding—that peak drive is difficult to reliably contain below a socially tolerable threshold following a very small release of organisms—appears robust to a variety of mitigating factors. Fundamentally, we exercise

caution by omitting application-specific phenomena that might aid containment in particular instances but not in general.

## Discussion

Our results suggest that current first-generation CRISPR-based gene drive systems for population alteration are capable of far-reaching—perhaps, for species distributed worldwide, global—spread, even for very small releases. A simple, constitutively expressed CRISPR nuclease and guide RNA cassette targeting the neutral site of insertion—an arrangement that could occur accidentally—may be capable of altering many populations of the target species depending on the homing efficiency of the organism in question. More generally, resistance can be problematic for intentional applications of gene drives, but we find that it is not a major impediment to invasion of unintended populations.

These findings raise two important questions: (1) How likely are unauthorized releases of self-propagating gene drive systems in the first place? (2) How likely are serious negative consequences given the apparently high likelihood of spread to most populations of the target species? Rigorously addressing these questions is an important direction for future work, and we can offer only opinions here. The answer to the first question likely depends on a large number of factors, such as species, application, containment strategies, economic motivations, drive development stages, geography, and the caution of the investigators, so we omit speculation here. However, we consider the answer to the second question to be clearer: although most laboratory gene drive systems are unlikely to cause ecological changes—they are typically predicted to be transient and are not designed to alter traits of the host organism, least of all interactions with other species—the history of genetic engineering offers many examples suggesting that substantial social backlash could be triggered by unauthorized spread of a self-propagating gene drive (*Funk and Rainie, 2015*; *Couzin and Kaiser, 2005*). Any such event could significantly reduce public support for interventions against diseases such as malaria that could possibly save millions of lives. We believe it would be profoundly unwise to proceed with anything less than an abundance of caution.

On a more technical note, our findings are specific to population alteration drive and cannot be directly generalized to self-propagating suppression drive, which could potentially self-extinguish before invading other populations. However, our results suggest a method for rough comparison between these scenarios: we find that the primary factor in determining drive spread between adjacent populations is the average number of migrants per generation (Analytic formulae for the escape probability in structured populations), which can, in principle, be compared between models. For example, an earlier model of suppression drive systems (*Deredec et al., 2011*) predicted a total number of drive-carrying organisms over time which is remarkably similar to our example of an inefficient alteration drive system that is rapidly outcompeted by resistant alleles (*Figure 1D*, middle). Thus, assuming comparable migration rates, it might not be surprising to see qualitatively similar levels of invasiveness. Accordingly, we urge researchers to exercise caution in developing or advocating for self-propagating suppression drives for applications other than malaria prevention—or similar projects intended to affect an entire species—until explicit models of invasiveness are available.

Additionally, our findings emphasize the importance of the containment strategy known as 'ecological confinement', which was proposed previously (*Esvelt et al., 2014*; *Akbari et al., 2015*). Given the risk that organisms may escape through accidents or outside intervention, laboratories in regions with endemic wild populations may wish to refrain from constructing self-propagating systems capable of invading those populations and undergoing unwanted spread. Laboratories in regions with endemic wild populations can reliably prevent accidental invasion by employing intrinsic molecular confinement mechanisms such as synthetic site targeting or split drive as recommended by the National Academies' report on gene drives (*National Academies of Sciences, Engineering, and Medicine, 2016*).

Perhaps most importantly, any development efforts looking ahead toward field trials, a component of the staged testing strategy outlined by the National Academies report, should be aware that there could be a high likelihood of unwanted spread across international borders, even from ostensibly isolated islands. The development of 'local', intrinsically self-exhausting gene drive systems (*Chen et al., 2007*; *Akbari et al., 2014*; *Noble et al., 2016*; *Magori and Gould, 2006*; *Gould et al., 2008*), sensitive methods of monitoring population genetics, and strategies for countering self-

propagating drive systems and removing all engineered genes from wild populations should be correspondingly high priorities.

## Materials and methods

### Well-mixed finite population model

To model gene drives in finite populations, we introduce a Moran-type model with sexual reproduction (illustrated in *Figure 1C*). We consider a population of $N$ individuals, each of which is diploid. We focus on a locus with three allelic classes: wild-type (W), CRISPR gene drive element (D) and drive-resistant (R). There are six possible genotypes: WW, WD, WR, DD, DR, and RR. We assign to each genotype $\alpha$ a reproductive rate $f_\alpha$.

The process proceeds in discrete time-steps, during each of which three events occur in succession (*Figure 1C*). First, two individuals are chosen without replacement for mating with probabilities proportional to their reproductive rates, so that genotype $\alpha$ is selected with probability

$$\frac{f_\alpha N_\alpha}{\sum_\beta f_\beta N_\beta}. \tag{1}$$

Here $N_\alpha$ is the number of individuals having genotype $\alpha$, and the sum in the denominator is over all six genotypes. Second, after selecting the two parents, the offspring genotype is chosen randomly based on the genotypes of the two parents. To proceed, we introduce notation $\alpha = AB$ to mean that genotype $\alpha$ consists of alleles $A$ and $B$, and we index these alleles via $\alpha_1 = A$ and $\alpha_2 = B$. Note that we track only one genotype for each heterozygote, implicitly combining counts for genotypes AB and BA. Using this notation, the probability that an offspring of genotype $\gamma$ is chosen given a mating between parents of genotypes $\alpha$ and $\beta$ is given by the quantity $q^\gamma_{\alpha\beta}$, which is equal to

$$\frac{q^{\gamma_1}_\alpha q^{\gamma_2}_\beta + q^{\gamma_2}_\alpha q^{\gamma_1}_\beta}{1 + \delta_{\gamma_1 \gamma_2}}. \tag{2}$$

Here $q^A_\alpha$ is a gamete production probability—the probability that a parent with genotype $\alpha$ produces a gamete with haplotype $A$—and $\delta_{AB}$ is the Kronecker delta, defined by $\delta_{AB} = 1$ if $A = B$ (i.e., if the offspring under consideration is a homozygote), and $\delta_{AB} = 0$ otherwise. The gamete production probabilities, $q^A_\alpha$, are determined by accounting for the gene drive process described above. They are given by: $q^W_{WW} = q^D_{DD} = q^R_{RR} = 1$, $q^D_{WD} = (1+P)/2$, $q^W_{WD} = (1-P)/2$, $q^W_{WR} = q^R_{WR} = q^D_{DR} = q^R_{DR} = 1/2$. The remaining values not listed, e.g., $q^R_{WW}$, are zero. Third, an individual is chosen uniformly at random for death. Thus, the population size remains constant. The resulting counts become the starting abundances for the next iteration of the process. The process is initialized with a small number, $i$, of drive homozygotes (DD) and the remaining population, $N - i$, wild-type homozygotes (WW). The process continues as described above either until a specified number of time steps have elapsed or until one of the three alleles has fixed. Any of the alleles can fix, but typically either the wild-type or resistant alleles fix, due to the emergence of resistance.

### Finite population model with population structure

To study the effects of population structure on drive containment, we extended the well-mixed model from the previous section. We now consider $l$ well-mixed subpopulations, each consisting initially of $N/l$ individuals. The process proceeds in discrete time steps, as before. In each time step, we either migrate an individual from one population to another, or we choose a particular subpopulation and proceed through one mating and replacement iteration, as outlined above. More specifically, one step of the process proceeds as follows (illustrated in *Figure 11*). With probability $m$, we initiate a migration event. In this case, we perform three steps. First, we choose a source population with probability proportional to its size. Second, we choose an individual uniformly at random from the source population for migration. Finally, we move the chosen individual to a linked subpopulation uniformly at random. Or, with probability $1 - m$, we initiate a mating event as described in the well-mixed section. To carry this out, we first choose the population in which the event will occur. We choose this population with probability proportional to the square of its total fitness, since this counts the rate of reproduction for every possible mating pair in the population (as matings occur

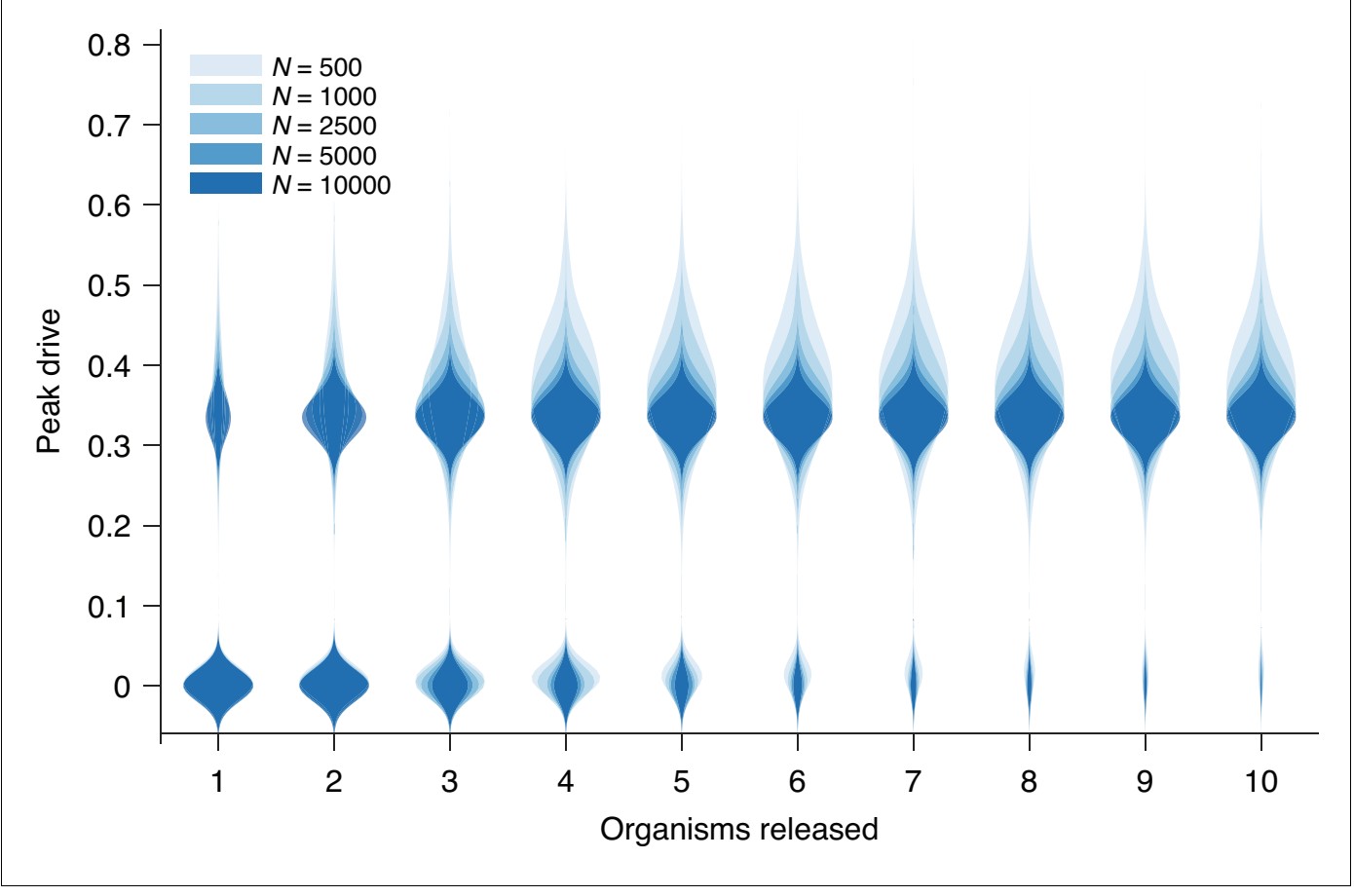

**Figure 3.** Peak drive distributions for variable release and population sizes. Parameters are chosen to correspond to *Figure 1E*: $P = 0.5$, $f = 0.9$ and neutral resistance. Population sizes are, from light to dark, $N = 500, 1000, 2500, 5000, 10000$. Note that $N = 500$ corresponds exactly to *Figure 1E*. Each distribution corresponds to $10^3$ simulations.

DOI: https://doi.org/10.7554/eLife.33423.005

with rates proportional to the fitness of each parent). We then step through one iteration of the well-mixed mating process within this subpopulation. Note that in this model the migration rate has a simple interpretation. The time between migrations is geometrically distributed with parameter $m$, so the mean time between migrations is $1/m$ time steps. Recall that a 'generation' is equal to the mean lifespan of an individual, that is, $N$ reproduction events or $N/(1-m)$ time steps. Then the typical time between migrations can be expressed with the units as generations:

$$\mathbb{E}[T] = \frac{1-m}{Nm}. \tag{3}$$

## Deterministic model

To compare our stochastic simulations with deterministic results, we use a recently published model (*Noble et al., 2017*). From that work, we employ the 'previous drive' model, as it was designed to agree with the existing proof-of-concept CRISPR drive constructs that we consider here. Specifically, we consider the case of 1 guide RNA ($n = 1$ in that work's notation), and zero production of costly resistant alleles ($\gamma = 1$).

## Population size

Above, we present results from simulations which assume populations of size $N = 500$. We claim that $N = 500$ is a reasonable approximation for the dynamics in the large-population limit, which is the

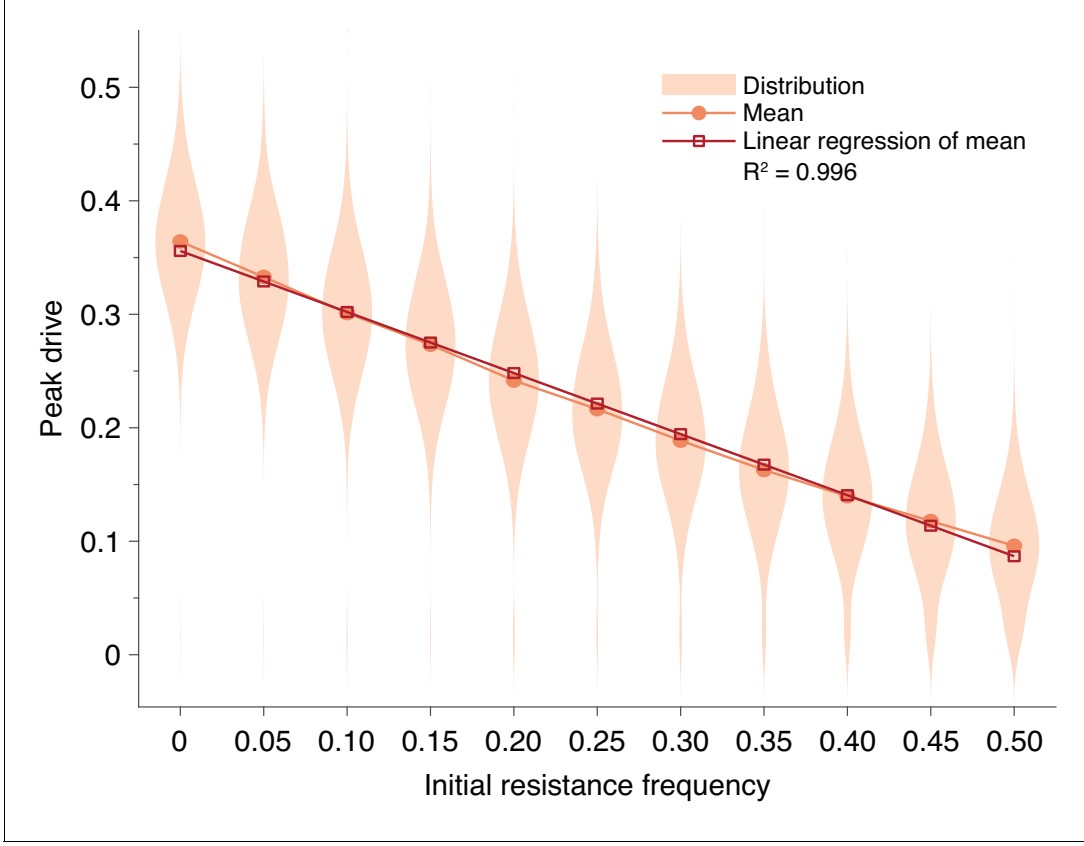

**Figure 4.** Pre-existing drive-resistant allele frequency linearly decreases peak drive. Distributions (violin plots), means (orange, circles) and linear regression of the mean values (red, squares). Parameters are chosen to correspond to *Figure 1E*: $P = 0.5$, $f = 0.9$, neutral resistance, $N = 500$. Each distribution corresponds to 5000 simulations.

DOI: https://doi.org/10.7554/eLife.33423.006

relevant regime for widespread invasion or for species with very large population sizes, e.g., mosquitoes. Here we briefly evaluate this claim.

*Figure 3* recreates *Figure 1E* from the main text with additional population sizes overlaid: $N = 1000$, $2500$, $5000$, and $10000$. The distributions narrow for larger $N$ until plateauing at roughly $N = 5000$. However, the central tendencies show little change with increasing $N$.

## Standing genetic variation

Several recent studies have explored the effect of pre-existing drive resistant alleles in a population brought about by standing genetic variation (SGV) at the target locus (*Unckless et al., 2017*; *Drury et al., 2017*). These studies developed deterministic models and showed that pre-existing resistant alleles—presumably neutral—should rapidly outcompete costly drives due to selection, resulting in rapid drive extinction. The study by Drury et al. (*Drury et al., 2017*) used sequencing to quantify this standing variation in diverse populations of flour beetles and found resistance-conferring mutations to exist at a wide range of frequencies, from $0$ to $0.375$, with an average of roughly $0.1$.

However, these studies were primarily concerned with long-term outcomes following drive release, in which case resistance certainly outcompetes the drive. For our purposes, however, we are concerned with the intermediate time regime in which the dynamics of resistance are less clear. Moreover, these studies employed deterministic models, whereas our model is stochastic. Here, we seek to understand the effect of SGV in our model.

To incorporate SGV, we simply alter the initial conditions: rather than introducing $i$ drive homozygotes into a population of $N - i$ wild-type homozygotes, we introduce $i$ drive homozygotes into a population consisting of $j$ resistant homozygotes (we choose resistant homozygotes for simplicity, since

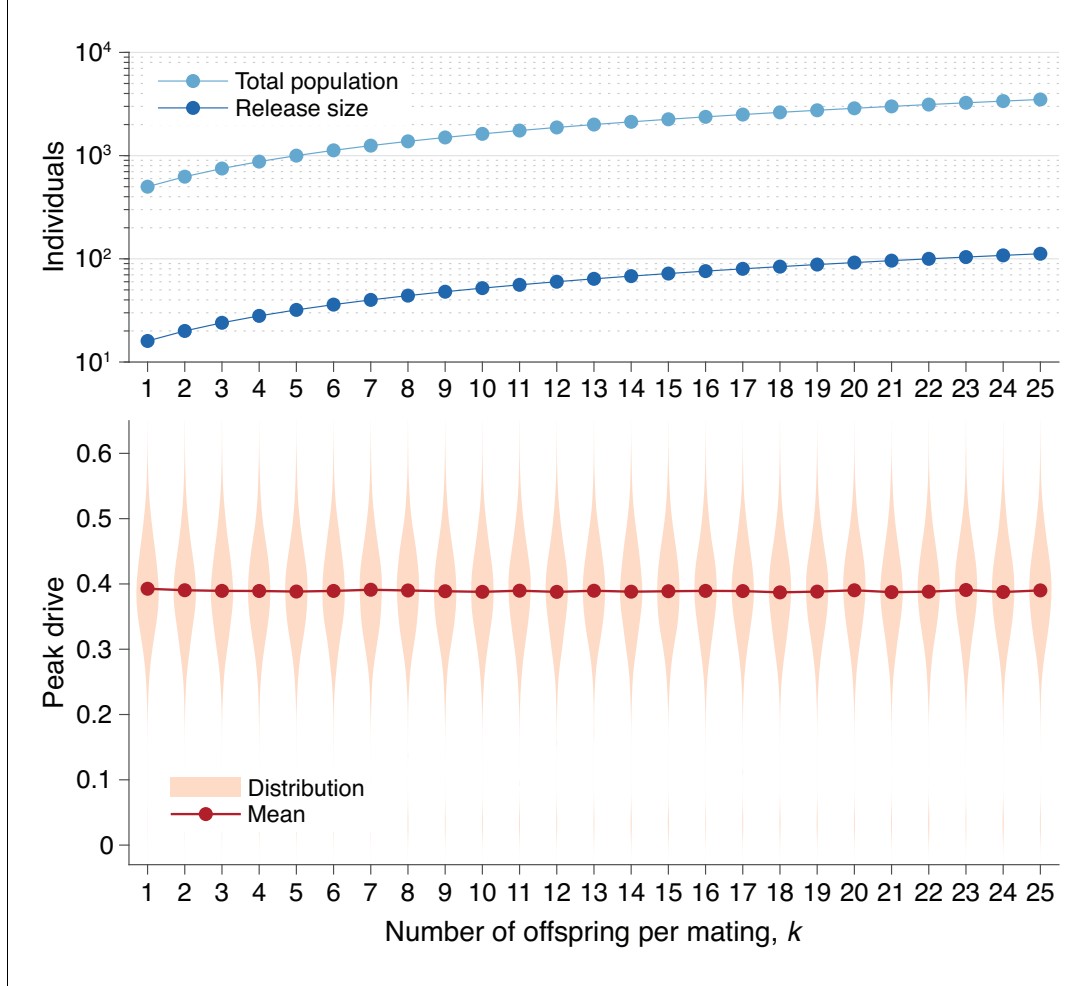

**Figure 5.** Peak drive distributions for varying numbers of offspring per mating with effective population and release sizes held constant. (top) Population and release sizes used in the simulations below. For the case $k = 1$, we use our usual population size of $N = 500$ with an initial release of $i = 16$ drive homozygotes. According to **Equation (5)**, the effective total population and release sizes in this case are $N_e = 250$ and $i_e = 8$. For other values of $k$, we use values of $N$ and $i$ which maintain constant effective population and release sizes: $N = N_e(2k + 6)/4$ and $i = i_e(2k + 6)/4$. These values are plotted: $N$ (light blue) and $i$ (dark blue). (bottom) Peak drive distributions assuming values of $N$ and $i$ as in the above plot. All employ $P = 0.5$, $f = 0.9$, and neutral resistance. Each distribution includes 5000 simulations.

DOI: https://doi.org/10.7554/eLife.33423.007

they rapidly go to Hardy-Weinberg equilibrium following release) and $N - i - j$ wild-type homozygotes. **Figure 4** shows the effect of SGV on peak drive for pre-existing resistance frequencies up to 0.5.

We find that the effect of SGV is to linearly decrease the mean peak drive ($R^2 = 0.996$). Our intuition for this result is as follows. Because the population is well-mixed, the effect of resistance is simply to decrease the size of the population that is susceptible to the effects of the drive. This can be roughly viewed as linearly scaling the drive-frequency axis. For example, if the population has a 0.1 frequency of resistant alleles immediately prior to release, then the population that is susceptible to drive is roughly 90% of the census population size, and the drive undergoes its usual dynamics within this subpopulation. There are of course complications to this simplistic explanation, e.g., selection increasing the size of the resistant population and diploidy mixing resistant and drive alleles. Furthermore, the linear relationship only holds for sufficiently low levels of SGV. In our example here, the relationship holds to roughly 0.5 initial resistance frequency. However, this is still higher than would be anticipated for drives engineered to spread in the wild.

Overall, our results suggest that a high level of SGV would be required to protect against drive invasion. In our conservative example (**Figure 4**) assuming 0.5 homing efficiency, 0.9 drive fitness,

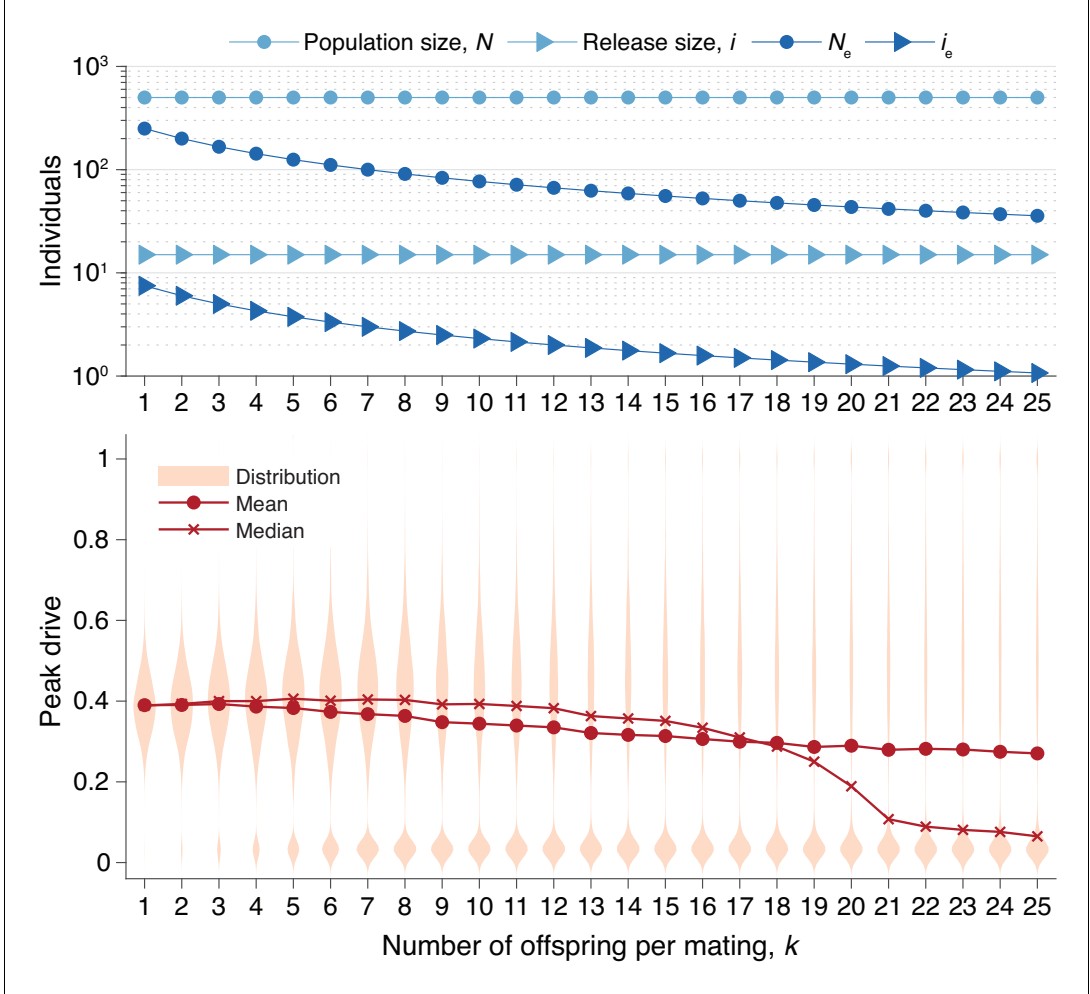

**Figure 6.** Peak drive distributions for varying numbers of offspring per mating with census population and actual release sizes held constant. (top) Population and release sizes used in the simulations below. Actual population size, $N$ (light blue, circles) and actual release size, $i$ (light blue, triangles). Note that $N = 500$ and $i = 15$ are constant. Effective values calculated via **Equation (5)**: population size, $N_e$ (dark blue, circles) and release size, $i_e$ (dark blue, triangles). (bottom) Peak drive distributions for simulations using indicated values of $k$ and population and release sizes as depicted above. Compare with **Figure 5** which holds the *effective* population and release sizes constant, whereas here we hold the *census* population and release sizes constant. All simulations employ $P = 0.5$, $f = 0.9$, and neutral resistance. Each distribution includes 5000 simulations.
DOI: https://doi.org/10.7554/eLife.33423.008

and neutral resistance, pre-existing resistance of greater than 0.5 frequency is required to contain peak drive to below 10% of the population, compared to 35% in the absence of SGV.

## Offspring number distribution

In the model presented above, we assume that each mating produces one offspring. However, a variety of application-relevant species are known to produce many offspring per mating. For example, female *Anopheles gambiae* mosquitoes can lay hundreds of eggs per lifetime (*Hammond et al., 2016*). It is not clear, *a priori*, how varying the offspring number distribution in our model would affect the results presented above. Thus we here analyze a simple extension of the model which allows us to vary the number of offspring following a given mating event.

To begin, recall our model. We consider a population of constant size $N$ with the following process: At each time-step, two individuals are chosen for mating; an offspring is sampled according to the parental genotypes; a third individual is chosen for removal from the population, and the parents' offspring takes its place. (We implicitly assume that these offspring are only the offspring which successfully reach adulthood, i.e., reproductive age). We now add a new parameter, $k$, which

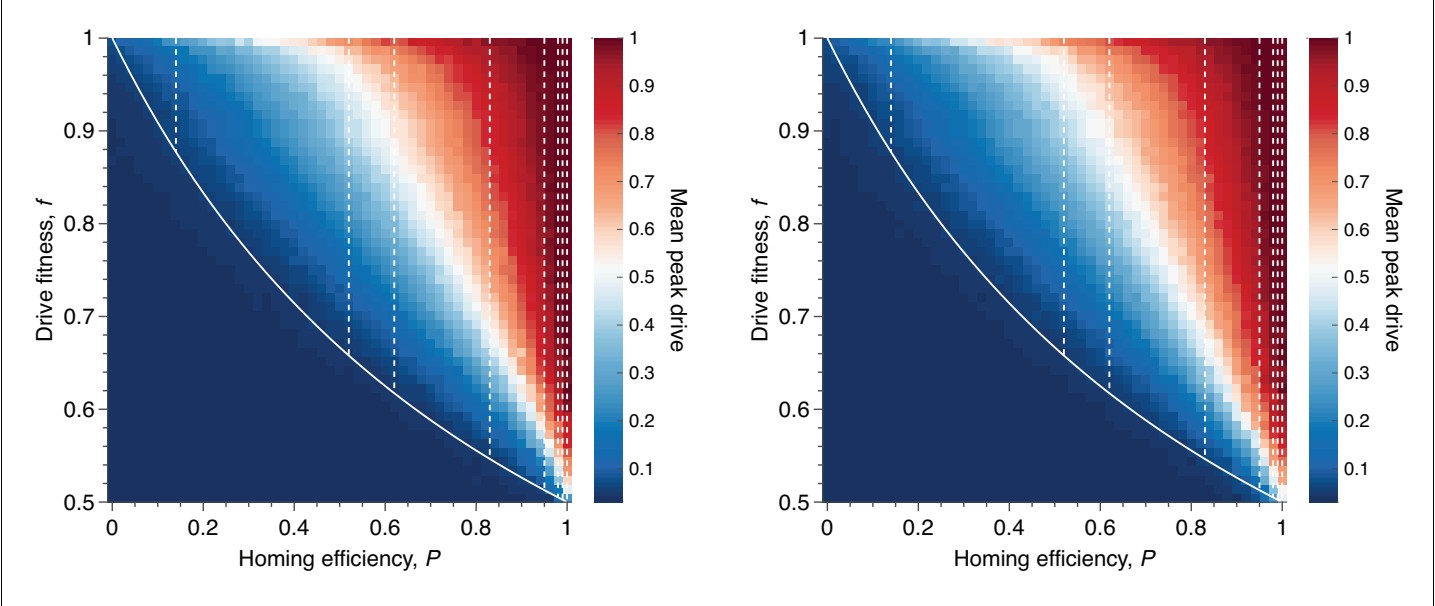

**Figure 7.** Mean peak drive for varying homing efficiency, *P*, and drive-individual fitness values, *f* (i.e., individuals with genotypes WD, DD, and DR), assuming that fitness affects birth rate (left) or death rate (right). The left panel corresponds to our standard model, shown in *Figure 1C*, while the right panel represents a modification: parents are chosen uniformly, and individuals die with probability proportional to the inverse of their fitness. The solid white line shows the boundary from *Figure 1B* indicating whether the drive is predicted to invade by deterministic models. The drive is only expected to invade based on deterministic models if the fitness/homing efficiency pair lie above the boundary. The dashed white lines indicate the empirically measured homing efficiencies from *Appendix 1—table 1* and *Figure 1B*. Each point in the grid (51 × 51) depicts an average of 100 simulations. Parameters used include a population size of 500, with an initial release of 15 drive homozygotes to ensure that trajectories establish. Neutral resistance is assumed throughout with no standing genetic variation.

DOI: https://doi.org/10.7554/eLife.33423.009

determines number of (adult) offspring produced by a mating pair. The process proceeds as before, except now *k* offspring are independently sampled from the parental genotypes, and *k* individuals are chosen uniformly (without replacement) for removal from the population. Clearly the model presented in the main text is the special case $k = 1$.

Note that this parameter *k* is not equivalent to brood size, clutch size, egg batch size, etc.—values often considered in the ecological literature—in that *k* describes the number of offspring produced per mating which successfully attain reproductive age. This number can of course be much lower than these other parameters due to death during juvenile life stages. We provide an example calculation for this parameter in *An. gambiae* at the end of this section.

We now argue that increasing the number of offspring per mating, *k*, corresponds to decreasing the effective size of the population, $N_e$. We omit rigorous proof here, but we provide a formula for the effective population size in our model and present numerical simulations as support. To begin, Hill showed in 1972 that the variance effective population size in the standard Moran model is (*Hill, 1972*)

$$N_e = \frac{4N}{2 + \sigma_X^2}. \tag{4}$$

Here *N* is the census population size, and $\sigma_X^2$ is the variance in the distribution of the total number of offspring produced by an individual over the course of its lifetime (*i.e.*, its lifetime reproductive success). It was proven that this formula holds both for the Wright-Fisher model with discrete generations and for the Moran model with overlapping generations, provided that $\sigma_X^2$ is the same and that the total number of individuals entering the population in each generation is equal (*Hill, 1972*). Our model meets both of these requirements—indeed, the only difference is that two parents are chosen to sample offspring types, rather than one, and this has no bearing on the number of offspring produced—so we conjecture that *Equation (4)* holds for our case as well.

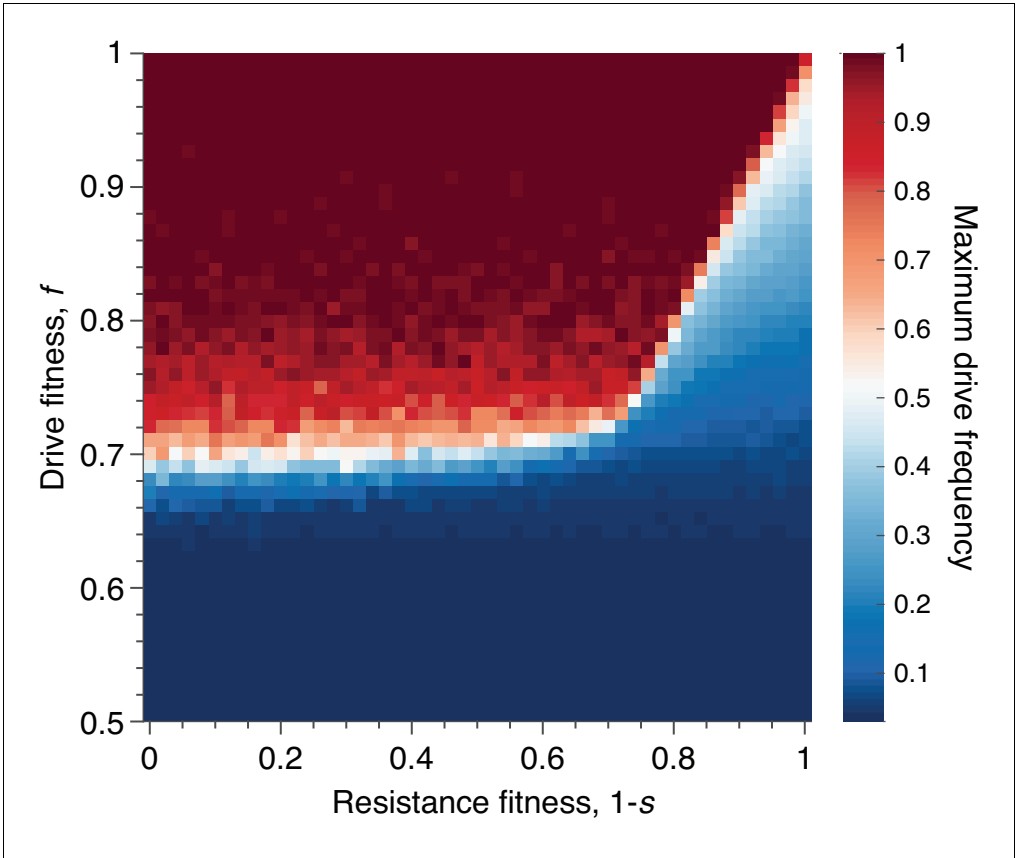

**Figure 8.** Mean peak drive for varying drive-individual fitness values, $f$, and resistant-individual (RR) fitness values, $1-s$, where $s$ is the cost associated with resistance. Each point in the grid ($51 \times 51$) depicts an average of 100 simulations. Parameters used include homing efficiency $P = 0.5$, population size of 500, with an initial release of 15 drive homozygotes to ensure that trajectories establish. Throughout we assume no standing genetic variation (i.e., the initial frequency of the resistant allele is 0).

DOI: https://doi.org/10.7554/eLife.33423.010

To proceed, we calculate $\sigma_X^2$ for our extended model and employ the variance effective population size given by **Equation (4)**. Consider one particular individual in the population, and let $t = 1, 2, \ldots$ count time-steps. As described, in each step, $k$ individuals are uniformly sampled (without replacement) for removal. Thus, an individual has probability $k/N$ of dying in each step. Its lifespan, $T$, is thus geometrically distributed, $T \sim \mathrm{Geometric}(k/N)$.

Next, let $X$ be a random variable describing the number of offspring an individual produces in its lifetime, so that $X|T$ is the number of such events given that the individual survives $T$ time-steps. Because each mating event is independent, $(X|T) \sim k \cdot \mathrm{Bin}(T, 2/N)$. The success probability derives from the fact that two individuals are chosen for mating in each time-step and that the process is neutral. Thus,

$$\mathbb{E}X = \mathbb{E}\mathbb{E}[X|T] = \mathbb{E}k(2/N)T = k(2/N)N/k = 2$$

and

$$\begin{aligned}
\mathrm{Var}(X) &= \mathbb{E}\mathrm{Var}(X \mid T) + \mathrm{Var}(\mathbb{E}(X \mid T)) \\
&= \mathbb{E}k^2 T(2/N)(1 - 2/N) + \mathrm{Var}(k(2/N)T) \\
&= kN(2/N)(1 - 2/N) + (2k/N)^2 N(N - k)/k^2 \\
&= 4 + 2k(N - 4)/N.
\end{aligned}$$

Returning to the variance effective population size expression in *Equation (4)*, we obtain for our model:

$$N_e = \frac{4N}{2k+6}. \tag{5}$$

Note that in the case $k = 1$ we recover $N_e = N/2$, which is the variance effective population size for the standard Moran model.

In *Figure 5*, we present peak drive distributions (as in *Figures 1E* and *3*) for varying values of $k$ with the effective population size, $N_e$, and effective release size, $i_e$, both determined by *Equation (5)*,

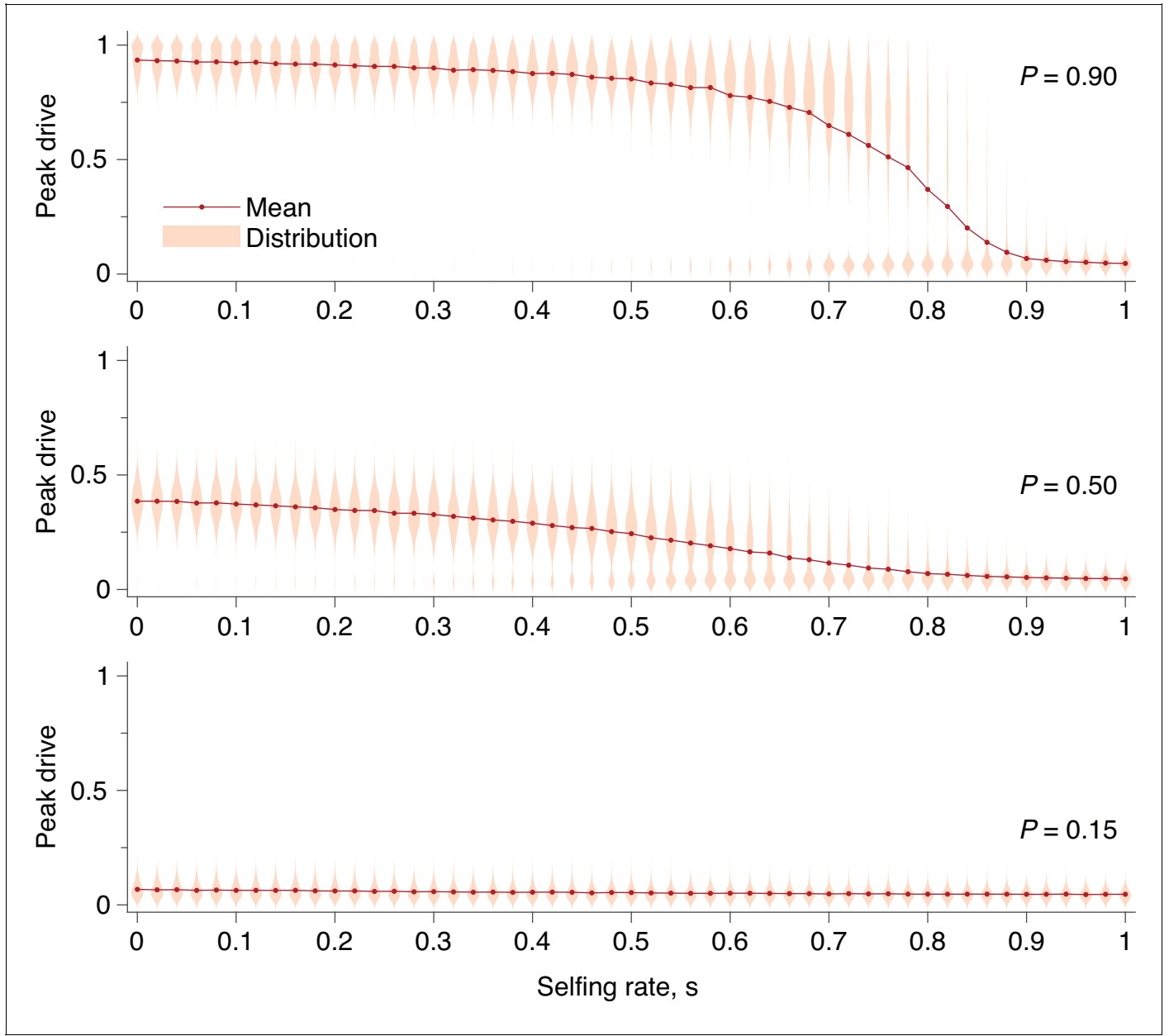

**Figure 9.** Peak drive distributions and means for varying selfing rates in our partial selfing model. (top) Effective drive, $P = 0.9$. (middle) conservative drive, $P = 0.5$, and (bottom) constitutive drive, $P = 0.15$. Each distribution comprises 1000 simulations. Parameters used include a population size of 500 with an initial release of 15 drive homozygotes. Neutral resistance is assumed throughout with no standing genetic variation, and the offspring number per mating is $k = 1$.

DOI: https://doi.org/10.7554/eLife.33423.011

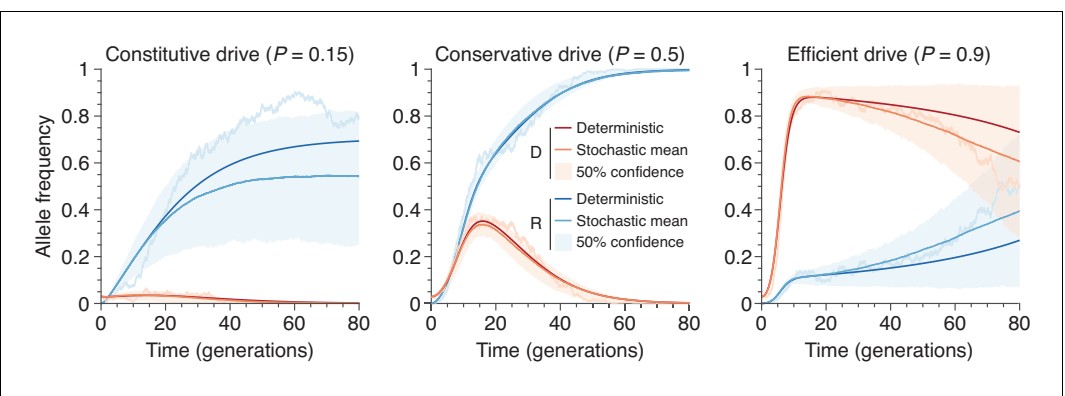

**Figure 10.** Finite-population simulations of 15 drive individuals released into a wild population of size 500, assuming low ($P = 0.5$) or high ($P = 0.9$) homing efficiencies, as well as a low-efficiency, constitutively active system ($P = 0.15$). Deterministic results (dark lines) and means of $10^3$ simulations (medium lines), individual sample simulations (light lines), and 50% confidence intervals (shaded). Drive frequencies red and resistant-allele frequencies blue.

DOI: https://doi.org/10.7554/eLife.33423.012

held constant. In this case we used $N_e = 250$ and $i_e = 8$, which correspond to $N = 500$ and an initial release of $i = 16$ in our standard model with $k = 1$. The peak drive distributions for all values of $k$ studied are approximately identical. This suggests that the dynamics for larger $k$ can indeed be inferred from the standard model with $k = 1$ and population/release sizes appropriately scaled via *Equation (5)*. An immediate consequence of this result is that releases of organisms which have many offspring (e.g., mosquitoes) are effectively smaller than would be expected from simply counting. For example, an organism which typically has 100 offspring that survive to adulthood would need a release size of roughly 258 to surpass the 10-individual initial release threshold we have observed. Note that the 10-individual threshold discussed throughout the text is the census release size; the effective release size is $i_e = 5$.

In *Figure 6*, we recalculate the distributions in *Figure 5* holding the *actual* population and release sizes constant, rather than their effective values. Two effects are apparent. First, the decrease in effective population size, $N_e$, leads to greater variation in peak drive among simulations that invade, *i.e.*, the distribution centered around $\approx 0.4$ widens. Second, the decrease in effective release size, $i_e$, leads to a greater probability of simulations immediately going extinct, *i.e.*, the relative mass of the mode centered around $\approx 0$ increases. In sufficiently large populations the first effect would be less pronounced—see *Figure 3*—while the second effect should apply for any small release.

Finally, as an example, we provide an estimate of our model's $k$ parameter for a particularly relevant species, *An. gambiae*. To do this, we find the typical size, $n$, of egg batches laid by females following a particular mating event; then we estimate the total number of these which survive to adulthood using parameters from the literature.

The first number, $n$, varies according to a variety of environmental and ecological factors (*Hammond et al., 2016*; *Yaro et al., 2006*), so we assume a large but reasonable value in order to avoid underestimating our parameter $k$. For this, we assume that $n \approx 186$, which is roughly the highest value observed by Hammond et al. in the CRISPR drive study (*Hammond et al., 2016*) and is in line with previous field work (*Yaro et al., 2006*).

To estimate the survival probability for each egg to adulthood, we employ the method and parameters presented by Deredec et al. (*Deredec et al., 2011*) Each egg goes through three juvenile stages before reaching adulthood—the egg stage, the larva stage, and the pupae stage. We denote the probabilities of surviving each of these stages by $\theta_0$, $\theta_L$, and $\theta_P$, respectively. The probability of a particular egg reaching adulthood is then $p = \theta_0 \theta_L \theta_P$. These parameters were estimated to be $\theta_0 = 0.831$, $\theta_L = 0.076$, and $\theta_P = 0.831$. Thus we have $p = 0.0525$.

Given this formulation, the number of eggs laid per mating event which reach adulthood is distributed according to $\mathrm{Bin}(n, p)$. We take the mean of this distribution to obtain:

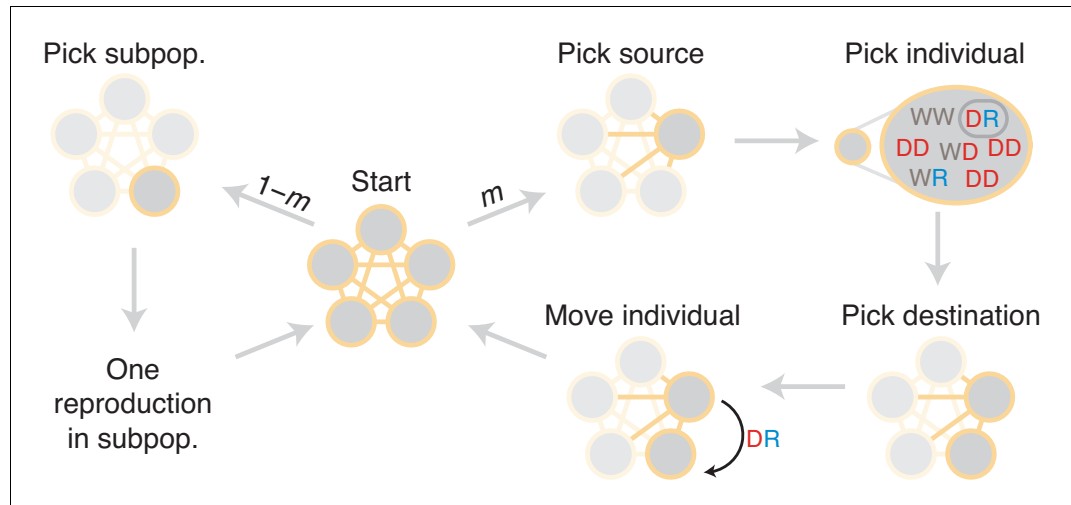

**Figure 11.** Diagram of simulation scheme. In each time step, a migration occurs with probability $m$, or a mating happens with probability $1 - m$. If a migration occurs, a source population is chosen randomly proportional to its size; an individual is chosen uniformly at random, then a destination is chosen uniformly at random, and the individual is moved. If a mating occurs, the dynamics proceed as in the well-mixed case for a particular subpopulation (*Figure 1C*).

DOI: https://doi.org/10.7554/eLife.33423.013

$$k \approx np = 9.76.$$

Therefore, while *An. gambiae* females exhibit large egg batch sizes, the value of $k$ for our model is much lower—indeed, low enough that the central tendency of the peak drive distribution remains roughly unchanged in *Figure 6*.

## Effect of varying fitness and homing efficiency

Above, we study various values of the homing efficiency, $P$, but we perform less exploration of the parameters governing drive fitness, $f$, and resistance cost, $s$. This is motivated primarily by the abundance of data for the former—see *Appendix 1—table 1*—and the lack of data for the latter parameters.

In addition, we have assumed throughout that death rates are identical for the various genotypes, while reproductive events occur with probabilities proportional to fitness. On the other hand, some drive constructs might behave the opposite way: reducing fitness by increasing an organism's death rate, while leaving its birth rate unchanged.

In this section we explore these three effects: (i) varying drive fitness across its entire range, (ii) varying the fitness cost of resistance across its entire range, and (iii) modifying the model so that death rates are affected by fitness, rather than birth rates.

To begin, we consider our standard model for fitness and study drive spread across the entire range of values for drive fitness, $f$, and homing efficiency, $P$. In particular, we consider 51 values of each parameter: $P \in [0, 1]$ and $f \in [0.5, 1]$, both evenly spaced, for a total of 2601 parameter pairs. For each pair, the average peak drive is calculated over 100 simulations, and the results are shown in *Figure 7*, left.

We find that maximum drive frequencies of greater than 0.3 are common across a wide range of drive fitness values. In particular, for our lower-bound estimate of empirical drive efficiency ($P = 0.5$), drives can confer fitness costs as high as $20\%$ before the peak drive drops below 0.3. For more typical empirical efficiencies ($P > 0.8$), the peak drive is typically greater than 0.5 even for costly drives ($f \approx 0.7$), and low-cost drives ($f > 0.9$) have peak drive of greater than 0.9.

We next modified our standard well-mixed model in the following way. Recall that the model involves choosing two parents to mate, then choosing an individual to die and be replaced by the parents' offspring. In our standard model, the two parents are chosen to reproduce with

probabilities proportional to their fitnesses, and an individual is chosen to die uniformly. In our modified model, we choose the two parents uniformly and then choose the individual to die with probability proportional to the inverse of its fitness. Results from the modified model are shown in *Figure 7*, right and are nearly identical to the results from the standard model.

In both cases, it is important to note that the peak drive and likelihood of invasion deemed socially acceptable for accidental release would likely be lower than those discussed above. With this in mind, our simulations suggest that if a drive is predicted to invade by deterministic models (i. e., if it lies above the boundary in *Figure 7*), then it will almost certainly reach a maximum frequency greater than $0.1$. While acceptable levels of peak drive are as-yet unknown and will likely vary between species, applications, jurisdictions and so on, spread to this extent will likely surpass it.

Finally, we sought to understand the effect of varying the fitness cost associated with drive-resistance. Throughout the text above we have assumed that resistance is neutral, as this presumably represents the best case for containment. However, drive constructs developed for applications are likely to employ resistance-mitigating strategies, such as multiplex targeting of essential genes (*Esvelt et al., 2014*; *Noble et al., 2017*), which essentially increase the fitness cost associated with drive-resistance. Thus, we ran simulations varying drive-individual fitness, $f$, in the range $f \in [0.5, 1]$, and resistant-individual (RR) fitness in the range $[0, 1]$, assuming conservative drive efficiency, $P = 0.5$. In both dimensions we considered 51 parameter values, evenly spaced, for a total of 2601 parameter pairs. For each pair, the average peak drive is calculated over 100 simulations, and the results are shown in *Figure 8*.

We find qualitatively that there are two regimes, determined by the fitness cost of resistance, $s$ (i. e., individuals with genotype RR have fitness $1 - s$), and the deterministic invasion condition, $f(1 + P) > 1$. In the figure, we assume that $P = 1/2$, so the deterministic invasion condition is simply $f > 2/3$. When the fitness cost of resistance, $s$, is sufficiently low ($s < 1/3$), then the dynamics are determined by the relationship between the fitness of drive individuals and the fitness of resistant individuals: if the fitness of drive individuals is greater than the fitness of resistant individuals, then the spread of the drive is dramatically improved—typically reaching fixation—compared to the baseline neutral-resistance case. However, if the fitness cost of resistance is sufficiently high ($s > 1/3$), then the improvement in drive spread brought about by increasing the cost of resistance saturates, since the drive can now be less costly than resistance ($f > 1 - s$) but also too costly to invade ($f < 2/3$). That is, for resistance costs higher than $1/3$, the mean peak drive as a function of drive fitness, $f$, remains essentially unchanged with increasing $s$, since the deterministic invasion condition can no longer be satisfied when the drive has fitness $f < 2/3$, no matter the cost of resistance.

## Inbreeding

Since the drive functions only in heterozygotes, inbreeding in a population—which in effect reduces the frequency of heterozygotes—would be expected to impact drive invasiveness. Indeed, this has been shown in recent theoretical studies by *Bull, 2017* and *Drury et al. (2017)* Thus we here extend our well-mixed model to include inbreeding and study its effect.

For simplicity, we consider a partial selfing model. In each update step of our process (see *Figure 1C*), we typically choose two parents for mating with probabilities proportional to their fitnesses. To include selfing, we instead choose the first parent as usual, with probability proportional to its fitness. We then choose the first parent as the second parent as well with probability $s$; or, with probability $1 - s$, we choose a second parent from the remaining population, with probability proportional to its fitness. Note that the fitness of each offspring is determined entirely by its genotype and does not account for inbreeding depression. Implicitly, we thus consider the case of zero inbreeding depression. As this effect helps protect against drive invasion, we essentially consider the worst-case scenario for drive containment (*Bull, 2017*).

Using our extended model, we then computed peak drive distributions for values of $s$ between 0 and 1 and for the three values of $P$ explored above: $P = 0.15, 0.5, 0.9$. The results are shown in *Figure 9*. We find that a fairly high degree of selfing is required to impact the peak drive distribution in a meaningful way. For highly effective drive, $P = 0.9$, the mass of the upper mode in the frequency distribution is larger than the lower mode until roughly $s \approx 0.75$. For conservative drive, $P = 0.5$, this occurs at roughly $s \approx 0.6$, and for ineffective drive there is little change, as the maximum frequency begins very near zero. To compare with previous results, we can consider the inbreeding coefficient rather than the selfing probability. In our model, the inbreeding coefficient, $F$, is given by $s/(2 - s)$.

Thus highly effective drive can tolerate inbreeding of $F \approx 0.6$ and conservative drive can tolerate $F \approx 0.43$.

## Comparison with deterministic model

To show that the deterministic ODE solutions provide reasonable approximations to the typical behavior of our stochastic model, we overlay numerical solutions to the ODEs for the systems studied in *Figure 1D* of the main text. The results are shown in *Figure 10*.

Throughout we have assumed that resistance is neutral with respect to the wild-type. This assumption is biologically realizable as resistance is conferred by changing sequence homology to the drive's gRNA—something that could be achieved with synonymous codon substitutions, for example. In practice, some resistance mutations could be costly and those that are neutral could be rare. However, assuming resistance is always neutral represents the worst-case scenario for drive invasiveness, as resistance can increase in frequency without being selected against with respect to the wild-type.

When resistance is no longer assumed to be neutral, other interesting dynamics can occur (*Traulsen and Reed, 2012*). In particular, when resistance is costly with respect to the wild-type, but not so costly as the drive and its cargo, the dynamics resemble the Rock-Paper-Scissors game. This allows the drive to avoid extinction indefinitely.

## Analytic formulae for the escape probability in structured populations

We consider a deme structured population, where each subpopulation has size $N$ and there are $n$ demes. We define a Moran-type process, where in each time step either a reproduction or migration event takes place (illustrated in *Figure 11*). A reproduction event occurs with probability $1 - m$ and a migration event occurs otherwise. If a reproduction occurs, then a subpopulation is selected proportional to the square of its total fitness. Next, two individuals in the subpopulation are selected proportional to their fitnesses and they produce an offspring according to the mechanism above. Finally, another individual from the subpopulation is chosen uniformly at random for death. If a migration event occurs, then an individual is selected uniformly at random and migrates to a new subpopulation uniformly at random. We denote the proportion of genotype $\alpha$ at time $t$ in the initial subpopulation by $P_t^\alpha$.

The process begins with $i$ drive homozygotes and $N - i$ wild-type homozygotes in a single subpopulation. The remaining subpopulations consist only of wild-type homozygotes. Let $E$ be the event that the frequency of drive alleles reaches 10% in a subpopulation other than where the drive was released, given that $i$ drive homozygotes were released in the initial subpopulation. We assume that $i$ is small with respect to $N$.

As an aside, note that the choice of 10% is arbitrary—any other percentage (less than the peak drive in the deterministic model, $c$) would be equivalent if $N$ is large enough. This is clear from *Figure 1E*, where either the drive does not invade and so peak drive is roughly equal to the initial frequency or the drive does invade and the peak drive is close to $c$. This claim is equivalent to stating that the probability that the drive starting at frequency $c_0$ attains frequency $c_1$ (such that $c_0 < c_1 < c$) before going extinct tends to 1. This behavior is typical of Moran-type models, since the extinction probability of $i$ drive homozygotes rapidly approaches 0, even in an infinite population, as $i$ increases (*Marshall, 2009*). Specifically, if we have $i = c_0 N$, then the extinction probability approaches 0 as $N$ becomes large, and moreover, if the drive does not go extinct, then it behaves almost deterministically and will reach frequency $c$ and thus also $c_1$.

Returning to approximating the probability of $E$, note that for $E$ to take place a drive allele has to migrate from the initial subpopulation *and* this allele has to survive stochastic fluctuations and avoid extinction in its new subpopulation. The drive alleles do not last indefinitely in the initial population. We denote the random time at which the drive alleles go extinct by $T$. As long as the initial drives do not go extinct due to stochastic fluctuations, the frequency of the drive increases rapidly, as it outcompetes the wild-type. Concurrently, resistant alleles are produced that eventually push the drive to extinction. This means that the drive has a finite time to migrate to other subpopulations. Although this process is stochastic it shows fairly deterministic behavior once there are a sufficient number of drive alleles (see *Figure 10*)—that is, if the drive avoids immediate extinction. Let $e_{i,j}$ be the probability that the drive survives stochastic fluctuations and avoids immediate extinction when

starting with $i$ drive homozygotes and $j$ heterozygotes. Implicitly, here we = assume that $e_{i,j}$ does not depend on whether the heterozygotes are wild-type or resistant heterozygotes. Note that when $i$ or $j$ are $\mathcal{O}(N)$, $e_{i,j}$ is approximately 1, so when $i,j \ll N$, we assume that the probability that the drive migrates is approximately 0. Moreover, since the drive will almost certainly go extinct, there is some time where the frequency of drive alleles is again much less than $\mathcal{O}(N)$. We also assume here that the probability that the drive migrates is approximately 0.

At each time step, there is a small probability that the drive migrates from the initial population and invades another subpopulation. To calculate, we first condition on the non-extinction of the initial $i$ drive homozygotes. Second, we note that if the drive does not migrate and avoid extinction in another subpopulation, then it does not do so at any particular time $t$. Third, we assume that these events for each $t$ are approximately independent. Finally, we numerically solve a deterministic ODE system representing the dynamics (*Noble et al., 2017*) to approximate the probability that the drive does not migrate at time $t$. Thus,

$$
\begin{aligned}
P\{E\} &= P\{\text{E}|\text{drive avoids extinction}\}e_{i,0} + P\{\text{E}|\text{drive does not avoid extinction}\}(1 - e_{i,0}) \\
&\approx P\{\text{E}|\text{drive avoids extinction}\}e_{i,0} \\
&\approx e_{i,0}\left(1 - \prod_{t=1}^{T} P\{\text{drive does not migrate and invade at time t}\}\right) \\
&= e_{i,0}\left(1 - \prod_{t=1}^{T} (1 - P\{\text{drive invades}|\text{drive migrates at time t}\}P\{\text{drive migrates at time t}\})\right) \\
&= e_{i,0}\left(1 - \prod_{t=1}^{T} \left(1 - m e_{1,0}\mathbb{E}P_t^{DD} - m e_{0,1}\left(\mathbb{E}P_t^{WD} + \mathbb{E}P_t^{DR}\right)\right)\right),
\end{aligned}
$$

since if the drive avoids extinction it will invade. Now we substitute the ODE solution $p_t^{\alpha\beta}$ for $\mathbb{E}P_t^{\alpha\beta}$ in the above expression to find that

$$
\begin{aligned}
P\{E\} &\approx e_{i,0}\left(1 - \exp\left(N\int_0^{T/(1-\lambda)} dt \log\left(1 - \lambda e_{1,0}p_{(1-\lambda)t}^{DD} - \lambda e_{0,1}\left(p_{(1-\lambda)t}^{WD} + p_{(1-\lambda)t}^{DR}\right)\right)\right)\right) \\
&\approx e_{i,0}\left(1 - \exp\left(\tfrac{N}{1-\lambda}\int_0^{T} dt \log\left(1 - \lambda e_{1,0}p_t^{DD} - \lambda e_{0,1}\left(p_t^{WD} + p_t^{DR}\right)\right)\right)\right).
\end{aligned}
$$

Here we approximated the product with an integral and used a change of variables.

Note that if $m = \mathcal{O}(1/T)$ and heuristically we replace $\mathbb{E}P_t^{\alpha}$ in the above expressions with its time average, denoted $\phi^{\alpha}$, then

$$
\begin{aligned}
&e_{i,0}\left[1 - \prod_{t=1}^{T}\left(1 - m e_{1,0}\mathbb{E}P_t^{DD} - m e_{0,1}\left(\mathbb{E}P_t^{WD} + \mathbb{E}P_t^{DR}\right)\right)\right] \\
&\approx e_{i,0}\left[1 - \left(1 - \tfrac{e_{1,0}\phi^{DD} + e_{0,1}\left(\phi^{WD} + \phi^{DR}\right)}{T}\right)^{T}\right] \\
&\approx e_{i,0}\left[1 - \exp\left(-e_{1,0}\phi^{DD} + e_{0,1}\left(\phi^{WD} + \phi^{DR}\right)\right)\right].
\end{aligned}
$$

Thus, when the migration rate is on the order of the inverse of the drive extinction time, the invasion probability is order 1.

## Acknowledgements

We thank J Wakeley for helpful discussions and M Edgington for helpful comments on the manuscript. CN received support from the NSF Graduate Research Fellowship Program under grant no. DGE1144152. KME was supported by the Burroughs Wellcome Fund (IRSA 73786). MAN, KME and GMC are funded in part by DARPA under the Safe Genes program.

## Additional information

### Funding

| Funder | Grant reference number | Author |
| --- | --- | --- |
| National Science Foundation | Graduate Research Fellowship, DGE1144152 | Charleston Noble |

| Burroughs Wellcome Fund | IRSA 73786 | Kevin M Esvelt |

The funders had no role in study design, data collection and interpretation, or the decision to submit the work for publication.

## Author contributions

Charleston Noble, Ben Adlam, Software, Formal analysis, Investigation, Visualization, Methodology, Writing—original draft, Writing—review and editing; George M Church, Supervision, Funding acquisition, Writing—review and editing; Kevin M Esvelt, Conceptualization, Supervision, Funding acquisition, Investigation, Methodology, Writing—original draft, Writing—review and editing; Martin A Nowak, Conceptualization, Formal analysis, Supervision, Funding acquisition, Methodology, Writing—review and editing

## Author ORCIDs

Charleston Noble  http://orcid.org/0000-0002-1253-1461
Kevin M Esvelt  https://orcid.org/0000-0001-8797-3945
Martin A Nowak  http://orcid.org/0000-0001-5489-0908

## Decision letter and Author response

Decision letter https://doi.org/10.7554/eLife.33423.020
Author response https://doi.org/10.7554/eLife.33423.021

## Additional files

### Supplementary files

• Transparent reporting form
DOI: https://doi.org/10.7554/eLife.33423.014

### Data availability

All empirical data reviewed for this study can be found in the empirical data supplement with detailed descriptions and references to their source studies. Data files, code to perform numerical simulations of all models presented (C++, Matlab), and code to reproduce all figures shown in the text (Matlab) can be found on GitHub at https://github.com/charlestonnoble/drive-invasiveness (copy archived at https://github.com/elifesciences-publications/drive-invasiveness).

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

## Appendix 1

DOI: https://doi.org/10.7554/eLife.33423.015

### Empirical data supplement

In *Appendix 1—table 1*, we present empirical homing efficiencies for all CRISPR gene drive constructs reported to date. These studies varied in multiple ways: they studied different organisms; they used different methods for counting drive constructs (ranging from direct genetic measurement, such as quantitative PCR, to indirectly observing visible phenotypes), and they sometimes observed differential inheritance rates between sexes, possibly due to differences in male and female gamete characteristics. Given this complexity, we elaborate here on the specific data we selected for review to produce *Appendix 1—table 1* and the reasoning for our choices.

**Appendix 1—table 1.** Empirical homing efficiencies for all CRISPR gene drive systems published to date. Details can be found in the Appendix.

| Organism | Ref. | System name | Efficiency |
|---|---|---|---|
| Yeast | (*DiCarlo et al., 2015*) | ade2::sgRNA | >99% |
| | | ade2::sgRNA + URA3 | 100% |
| | | sgRNA + ABD1 | 100% |
| | | cas9 + sgRNA | >99% |
| | | ADE2 + sgRNA + cas9 | >99% |
| Fruit flies | (*Gantz and Bier, 2015*) | $\gamma$-MCR | 97% |
| | (*Champer et al., 2017*) | nanos | 62% |
| | | vasa | 52% |
| | | additional nanos | 40–62% |
| | | additional vasa | 37–53% |
| Mosquitoes | (*Gantz et al., 2015*) | AsMCRkh2 (male) | 98% |
| | | AsMCRkh2 (female) | 14% |
| | (*Hammond et al., 2016*) | AGAP011377 | 83% |
| | | AGAP005958 | 95% |
| | | AGAP007280 | 99% |

DOI: https://doi.org/10.7554/eLife.33423.016

To begin, all studies performed some variation of producing drive/wild-type heterozygotes (DW), followed by counting the number which converted their wild-type allele to a drive allele. There were two main approaches.

1. Some constructs acted in the early embryo, in which case WW and DD individuals were mated to produce offspring which were initially WD. Observations were then made of adult genotypes. DD individuals must have undergone drive conversion, while WD individuals must have avoided conversion. Without drive, all adults are expected to be WD, but with drive, all are expected to be DD.
2. Other constructs acted in the germline of adults, so that adult WD individuals produce D gametes more often than chance under the effects of drive. To study these constructs, WD individuals were mated with WW individuals. Without drive, half of adults should be WW, and half should be WD. With drive, however, all adults should be WD.

To employ a consistent strategy across the studies, we calculate two numbers for each drive construct: (i) the total number of initial alleles counted which were drives or were subject to drive, $T$, and (ii) the total number of resulting drive alleles, $D$. The homing efficiency can then be calculated in the following way:

$$P = \frac{2D}{T} - 1$$

Notice that if drive is perfectly efficient ($P = 1$), we have $D/T = 1$, *i.e.*, there are twice as many drive alleles as starting heterozygotes, while under standard inheritance ($P = 0$), the number of drive alleles is unchanged from the initial heterozygous state, $D/T = 1/2$. Below, we explain our calculations of these quantities for *Appendix 1—table 1*.

### Yeast, *DiCarlo et al. (2015)*

The study by DiCarlo et al. studied five distinct gene drive systems in yeast (*DiCarlo et al., 2015*). We address each distinct system in subsections below.

### ade2::sgRNA

This is the basic split drive system containing only a guide RNA. Its design is depicted in *Figure 2B*, and it is described on pp. 1250–1251, with results pictured in *Figure 2D* and *Figure 4*. Drive abundances were measured via colony counting (*Figure 2D*), obtaining absolute colony numbers, and via qPCR (*Figure 4*), obtaining relative abundances of drive alleles. By the colony counting method, the drive efficiency is measured at 100% ($D = T = 72$). By the qPCR method, >99% of alleles counted from offspring were drive alleles, so $D > 0.99T$. Therefore:

$$P > 0.99$$

Strictly speaking, the inequality $D > 0.99T$ entails $P > 0.98$, but we set this to $P > 0.99$ because the qPCR results were indistinguishable from $100\%$. We make a similar approximation below for systems 4 and 5.

### ade2::sgRNA +URA3

This system aimed to test whether an associated 'cargo' gene could be spread with the minimal drive element. Its design is depicted in *Figure 3a*, and results are shown in *Figure 3b*. The related experiment measured drive presence via a visible phenotype (red pigment). In total, 60 haploids were red, or $D = 60$, out of 60 total alleles, $T = 60$. Thus:

$$P = 1$$

### sgRNA+ABD1

The sgRNA +ABD1 drive system tested the ability to target a recoded essential gene. Its design is depicted in *Figure 3c*, and results are discussed in the text (first full paragraph on pp. 1252). The presence of the drive was measured via sequencing of the ABD1 locus. In total, 72 haploids were found to have the drive, $D = 72$, out of $72$ counted, $T = 72$.

$$P = 1$$

### cas9+sgRNA

The first example of an 'autonomous' drive in the paper, this system is depicted in *Figure 5a*. It consisted of a gRNA and cas9 together targeting the ADE2 locus (recoded due to safety/containment considerations). The fractional abundance of drive allele was measured by performing qPCR on diploid offspring from wild-type/drive haploid matings; the corresponding data is found in *Figure 5b*. The fractional abundance of the drive allele was measured to be >99%, so $P > 0.99$, as for the first construct above.

$$P > 0.99$$

### ADE2+sgRNA + cas9

This system is DiCarlo et al.'s example of a 'reversal' drive, designed to target and overwrite the autonomous drive (cas9 +sgRNA, directly above). The system is depicted in *Figure 5c*.

The drive efficiency was measured in the same way as that for the cas9 +sgRNA drive (qPCR to calculate fractional abundance of the overwriting drive allele in diploid offspring from haploid matings). The fractional abundance was calculated to be >99%, so $P>0.99$, as above.

$$P>0.99$$

### Fruit flies, *Gantz and Bier (2015)*

Gantz et al. constructed an X-linked drive construct targeting the (X-linked) *yellow* locus in *Drosophila melanogaster* and acting in the early embryo (**Gantz and Bier, 2015**). The drive functions to knock out the *yellow* gene, which produces a yellow-body phenotype, denoted $y-$, due to lack of black melanin pigment formation. The wild-type phenotype is referred to as $y+$. Females with <2 copies of the drive or males with 0 copies should appear $y+$, while females with 2 copies of the drive or males with one copy should appear $y-$. The related data is found in **Figure 2E** and Table 1.

Two sets of crosses were performed: (i) drive-males with wild-type females, and (ii) drive-females with wild-type males. To tabulate the allele counts $D$ and $T$, we discuss the two crosses separately.

First, cross (i): In this cross, male offspring could not have possibly inherited a drive allele nor received one through conversion. This is because the only allele they could have inherited from the drive-male parent was the Y chromosome, but the drive is X-linked. Thus we do not consider male offspring in the total. As for female offspring, these should inherit exactly one drive allele and one wild-type allele prior to conversion. Then the adult female individuals should appear $y-$ if and only if drive-mediated conversion was successful. Thus we add exactly two alleles for each female offspring toward the total allele count, while we add one or two drive alleles to the drive allele count if the adults are $y+$ or $y-$, respectively. This yields $D_m = 40 \times 2 + 1 \times 1 = 81$ and $T_m = 40 \times 2 + 1 \times 2 = 82$. The drive efficiency for this cross is $P_m = 2D_m/T_m - 1 = 0.976$.

Second, cross (ii): In this cross, male offspring are again uninformative, since each should inherit exactly one drive allele from the female parent and one Y allele from the male wild-type parent. Thus we ignore male offspring in our counting. Female offspring, on the other hand, should all begin as WD embryos, with $y+$ phenotypes. Then adults are $y-$ if and only if they have undergone drive-mediated conversion. Thus we count two alleles for every female offspring in the total, one drive allele per $y+$ adult and two drive alleles per $y-$ adult. This yields $D_f = 203 \times 2 + 1 \times 6 = 412$, and $T_f = 203 \times 2 + 6 \times 2 = 418$. The drive efficiency for this cross is thus $P_f = 2D_f/T_f = 0.971$.

We then consider crosses (i) and (ii) together to calculate the overall drive efficiency. This yields:

$$P = 2\frac{D_m + D_f}{T_m + T_f} - 1 = 2\frac{81 + 412}{82 + 418} - 1 = 0.972$$

### Fruit flies, *Champer et al. (2017)*

Champer et al. constructed two CRISPR gene drive constructs in *D. melanogaster* (**Champer et al., 2017**). The first resembled the *vasa* promoter-driven construct from Gantz et al., discussed in the section immediately above. An important addition, however, was a DsRed fluorescent protein as payload in the drive construct, which allows the drive to be detected in heterozygotes, as its red fluorescent phenotype is dominant. The second construct used the *nanos* promoter, which has been shown to restrict drive function to the germline and is expected to produce less toxicity (and thus a lower fitness cost associated with the drive construct).

#### *vasa* construct

This construct was similar to the one studied by Gantz et al., discussed above. The construct targets the X-linked *yellow* gene. Disruption of the gene produces a recessive yellow

phenotype, while the drive itself carries a DsRed payload, producing a dominant red fluorescent eye phenotype. To assess the construct's homing efficiency, wild-type males were crossed with heterozygous DW females. In this setup, all progeny should exhibit the red eye phenotype if the drive is perfectly efficient, while roughly 50% of progeny should exhibit the red eye phenotype in the absence of conversion. Here we count toward the total number of drive or susceptible alleles one allele per male offspring and one allele per female offspring, since in either case only one allele is inherited from the drive parent. Toward the number of drive alleles, we count one per offspring if the offspring displays the DsRed phenotype and zero otherwise. This data is shown in Table 2B of the *Champer et al., 2017* study. We count as follows: $D_f = 909 + 4 = 913$ (i.e., the number of drive alleles counted over female offspring), $T_f = 909 + 4 + 316 = 1229$, $D_m = 953$, $T_m = 953 + 265 + 3 = 1221$. Then we obtain:

$$P = 2\frac{D_m + D_f}{T_m + T_f} - 1 = 2\frac{953 + 913}{1221 + 1229} - 1 = 0.523.$$

### *nanos* construct

This construct is essentially the same as the *vasa* construct, except that it uses a different promoter and targets a different sequence in the *yellow* gene (the coding sequence, rather than the promoter as in the previous construct). The data is found in Table 1B of the *Champer et al., 2017* study. We count potential drive alleles and total alleles as above. Our count is as follows: $D_f = 290 + 100 + 108 = 498$, $T_f = 290 + 100 + 108 + 119 + 10 + 9 = 636$, $D_m = 594$, $T_m = 594 + 11 + 103 + 2 = 710$. We obtain:

$$P = 2\frac{D_m + D_f}{T_m + T_f} - 1 = 2\frac{594 + 498}{710 + 636} - 1 = 0.622.$$

### Additional data

The constructs described above were then tested in a variety of additional *D. melanogaster* lines, detailed in Table 3 of that work. The authors' efficiency calculations are detailed in the S1 Dataset. For the vasa construct (two lines), the minimum is $P = 0.37$, and the maximum is $P = 0.53$. For the nanos construct (seven lines), the minimum is $P = 0.40$, and the maximum is $P = 0.62$.

## Mosquitoes, *Gantz et al. (2015)*

In this study, Gantz et al. constructed an autonomous CRISPR-based gene drive system in the malaria vector mosquito *Anopheles stephensi* (*Gantz et al., 2015*). The construct comprises two effector genes with anti-*Plasmodium falciparum* activity, a dominant marker gene (DsRed), and the CRISPR components (Cas9 with a single gRNA), spanning roughly 17 kb. The construct targets the *kynurenine hydroxylase*^*white* ($kh^w$) locus, which has a recessive white-eye phenotype. The effect of this targeting is that drive/wild-type heterozygotes display a DsRed phenotype, while drive homozygotes display both DsRed and white eyes.

While this one construct was made and studied, it exhibited differential transmission between lines founded by drive males/wild-type females and drive-females/wild-type males. More specifically, lines in which drive alleles are inherited only through male parents display drastically higher drive efficiencies than lines in which the drive allele is inherited at some point via a female parent. To explain this discrepancy, the authors propose a model whereby in crosses between transgenic females and wild-type males, maternal deposition of Cas9 in eggs results in NHEJ-mediated disruption of the paternally derived wild-type chromosome in the early embryo. Crosses between transgenic males and wild-type females, on the other hand, do not see Cas9 deposited in the early embryo, and Cas9 cutting is better contained to the later germline, where HDR is more efficient.

To account for this discrepancy, we choose to consider these two cases separately and report homing efficiencies for each.

## Transgenic male lines

Here we consider all offspring (larvae + adults) whose drive alleles (or potentially-inherited drive alleles) have been passed down only through male ancestors. This includes all offspring from the male-founder crosses in Table 1 of the main text (10.1 $G_2$ and 10.2 $G_2$), as well as crosses 6 and 8 in Table 2 (also **Figure 3**). We choose to compile all alleles from each of these crosses together to calculate an average efficiency across all available data. Because the constructs are on autosomes, we treat male offspring and female offspring identically, and we count toward the total allele count, $T$, one allele from each offspring (since at most one drive allele can be inherited in each cross), and we count toward the drive allele total, $D$, one allele for each DsRed$^+$ individual observed, since this is a dominant marker for the drive. Finally, we consider both larvae and adults identically, as conversion is anticipated to have occurred before this stage, and results are similar between adults and larvae. Values of $D$ and $T$ for each cross are displayed in **Appendix 1—table 2**.

To obtain an average efficiency for the construct, we sum the values of $D$ and $T$ across all crosses in **Appendix 1—table 2**. We obtain:

$$P = 2\frac{8985}{9081} - 1 = 0.979.$$

**Appendix 1—table 2.** Gantz et al., *An. stephensi* transgenic male lines. (left) Phenotypes of $G_3$ progeny. (right) Phenotypes of $G_4$ progeny.

| $G_3$ crosses | $D$ | $T$ | Reference |
|---|---|---|---|
| 10.1 $G_2 \times$WT, larval | 829 | 832 | Table S3 |
| 10.2 $G_2 \times$WT, larval | 3060 | 3085 | Table S4 |
| 10.1 $G_2 \times$WT, adult | 833 | 836 | Table S5 |
| 10.2 $G_2 \times$WT, adult | 1258 | 1274 | Table S6 |
| Total | 5980 | 6027 | — |
| **$G_4$ crosses** | **$D$** | **$T$** | **Reference** |
| Cross 6, larval | 949 | 955 | Table S7 |
| Cross 8, larval | 609 | 628 | Table S8 |
| Cross 6, adult | 882 | 888 | Table S10 |
| Cross 8, adult | 565 | 583 | Table S11 |
| Total | 3005 | 3054 | — |

DOI: https://doi.org/10.7554/eLife.33423.017

## Transgenic female lines

To understand the effect of maternal Cas9 deposition, we count all offspring (larvae + adults) from crosses such that the any (potentially) inherited drive allele has been inherited via a female parent at least once. This includes no $G_3$ offspring, as the drive alleles present in $G_2$ parents were inherited from $G_1$ males. Thus we include only $G_4$ offspring of $G_3$ parents, specifically Crosses 1–4, and as for the transgenic male lines, we sum both larval and adult crosses. Values of $D$ and $T$ for each cross are displayed in **Appendix 1—table 3**. Summing the values in **Appendix 1—table 3** yields:

$$P = 2\frac{2860}{5000} - 1 = 0.144.$$

**Appendix 1—table 3.** Gantz et al., *An. stephensi* transgenic male lines. (left) Phenotypes of $G_4$ larvae. (right) Phenotypes of $G_4$ adults.

| $G_4$ larvae | $D$ | $T$ | Reference |
|---|---|---|---|

*Appendix 1—table 3 continued on next page*

*Appendix 1—table 3 continued*

| G$_4$ larvae | D | T | Reference |
|---|---|---|---|
| Cross 1 | 28 | 48 | Table S7 |
| Cross 2 | 332 | 635 | Table S7 |
| Cross 3 | 204 | 324 | Table S8 |
| Cross 4 | 372 | 632 | Table S8 |
| Total | 936 | 1639 | — |
| **G$_4$ adults** | **D** | **T** | **Reference** |
| Cross 1 | 19 | 35 | Table S10 |
| Cross 2 | 306 | 554 | Table S10 |
| Cross 3 | 169 | 272 | Table S11 |
| Cross 4 | 1430 | 2500 | Table S11 |
| Total | 1924 | 3361 | — |

DOI: https://doi.org/10.7554/eLife.33423.018

## Mosquitoes, *Hammond et al. (2016)*

In this study, the authors construct three CRISPR-based gene drive systems in the malaria vector *An. gambiae*, each targeting a different gene with a recessive female sterility phenotype upon disruption (*Hammond et al., 2016*). These are examples of suppression drives whose purpose is to reduce or eradicate wild populations. Each drive construct carries a copy of Cas9, a single guide RNA, and red fluorescent protein (RFP) which has a dominant fluorescent phenotype. Each construct targets one of three female fertility genes, referred to as AGAP011377, AGAP005958, and AGAP007280, but otherwise they are identical.

To determine homing efficiency, drive-heterozygotes were crossed with wild-type homozygotes, and offspring were scored visually for the presence of the dominant marker RFP gene. Thus in our tabulations, we count one allele per individual toward the total, $T$, and we count one allele per RFP$^+$ individual toward the drive allele count, $D$. Furthermore, the outcrosses were performed over several generations. To obtain average homing efficiencies, we sum drive alleles and total alleles over G$_2$, G$_3$, G$_4$, and G$_5$ generations, when applicable. (Some constructs were tested over more generations than others.) This data is found in Table 2 in the study. Furthermore, we sum across male- and female-drive parent crosses, since we would expect these to behave identically with respect to homing, given that the female drive parents are capable of producing offspring.

## AGAP011377

This construct was studied over generations G$_2$ to G$_5$ in Table 2. The total number of relevant alleles resulting from crosses between drive-male parents and wild-type females was $T_m = 636 + 1631 + 1654 + 505 = 4426$, while the male drive total was $D_m = 581 + 1442 + 1550 + 491 = 4064$. The female total was $T_f = 60 + 92 + 142 = 294$, and the female drive total was $D_f = 55 + 70 + 121 = 246$. The average efficiency is then:

$$P = 2\frac{D_m + D_f}{T_m + T_f} - 1 = 2\frac{4064 + 246}{4426 + 294} - 1 = 0.826.$$

## AGAP005958

This construct was studied over generations G$_2$ and G$_3$. There were no offspring from female-drive crosses to wild-type due to the low fertility of these individuals. The total was $T = 1689 + 278 = 1967$, and the drive total was $D = 1654 + 268 = 1922$. The efficiency is thus:

$$P = 2\frac{D}{T} - 1 = 2\frac{1922}{1967} - 1 = 0.954.$$

### AGAP007280

This construct was studied over generations $G_2$ and $G_3$. The male total was $T_m = 1383 + 505 = 1888$, and the male drive total was $D_m = 1377 + 499 = 1876$. The female total was $T_f = 257$, and the female drive total was $D_f = 255$. The efficiency is:

$$P = 2\frac{D_m + D_f}{T_m + T_f} - 1 = 2\frac{1876 + 255}{1888 + 257} - 1 = 0.987.$$

