## [Decision Letter]

Thank you for submitting your article "Current CRISPR gene drive systems are likely to be highly invasive in wild populations" for consideration by *eLife*. Your article has been reviewed by three peer reviewers, one of whom is a member of our Board of Reviewing Editors, and the evaluation has been overseen by Diethard Tautz as the Senior Editor. The following individuals involved in review of your submission have agreed to reveal their identity: James Bull (Reviewer #2); Bernard Dujon (Reviewer #3).

The reviewers have discussed the reviews with one another and the Reviewing Editor has drafted this decision to help you prepare a revised submission.

Your paper shows that although resistance prevents drive systems from spreading to fixation in large populations, even very ineffective systems are highly invasive. Based on this it is argued that standard gene drive systems should not be developed nor field-tested in regions harboring the host organism.

The main concern of the reviewers is that the rather forceful message of the paper is not really substantiated in the sense that it is unclear why a high peak frequency of gene drive systems is necessarily a bad thing. For example, transposable elements have recently risen to high frequency in wild *Drosophila* populations, and while investigating the mechanism of this rise (and its potential prevention) are very interesting topics, it is unclear that this rise is a bad thing for these *Drosophila* populations. In other words, a rapid spread does not necessarily imply doom. I think this should be acknowledged in some form by the authors, and the message should be tuned accordingly.

The reviewers also raised a number of technical points, e.g. about the effects of a cost of resistance. These concerns should be carefully addressed.

*Reviewer #1:*

This is an interesting paper on the dynamics of the CRISPR gene drive system. Contrary to prevalent opinion, the authors conclude that invading gene drive systems may have a lasting impact even if they do not go to fixation. In particular, substantial "peak drive" occurs under many conditions. Based on this the authors caution against being careless with such systems, particularly when it comes to introduction of gene drive systems into wild populations.

The theoretical analysis is comprehensive and appears to be sound. The conclusions regarding the danger of gene drive may be a bit too sweeping.

Specific comments:

- The authors start out by pointing out the need for stochastic models of populations with finite size. However, in Figure 9 they effectively show that deterministic models make the same predictions as their stochastic models (if I understand this figure correctly). So why do we need stochastic models after all? This should be better explained, since it is part of the main rationale for the analysis presented in the paper.

- The model assumes that death is random, but births occur according to fitness. What if birth is random, and death is proportional to fitness? (There are some models in which this makes a difference…)

- It seems that in the model, invasion is always initialized with homozygotes? Can this assumption be relaxed?

- I don't really understand this statement, "Invasion is very unlikely when the drive is not initially favoured by selection." I thought the drive allele is never favoured by selection, as measured by the fitness values.

- Why show mean and median in Figure 1E,F.

- I don't really understand Figure 2E: it shows that the probability of invasion into one local population depends on migration rate, but that probably should not depend on migration at all, because as I understand it, the figure presumably refers to invasion of the drive in the local population into which the drive is initially released.

- Choosing populations proportional to the "square of total fitness" seems odd. Shouldn't it be the sum of squares of individual fitness values, or something like that?

*Reviewer #2:*

Conceptually, I view this paper as having two parts. The first is an analysis of gene drive models to study the spread of drives and consequent evolution of resistance to the drives. The second part is a value judgement on the deployment of drives. The latter occupies little space in the paper, but is extreme and certain to attract attention ("Contrary to the National Academies report on gene drive, our results suggest that standard drive systems should not be developed nor field-tested in regions harboring the host organism" is the last sentence of the Abstract). For convenience, I will refer to these as Part I and II, even though the paper is not structured that way.

Part I is possibly the most comprehensive analysis to date of drive and resistance evolution under alternative population structure scenarios. The overall message is that drives are highly invasive, albeit only temporary, despite resistance evolution. The quantitative details are not intuitive, but I would say the qualitative conclusion is obvious. Most of the models here assume a drive fitness cost of 10%, and what is not obvious is now far a drive allele would rise before being shut down by resistance evolution – the resistance evolution is assured by the fitness cost of the drive. So one cannot immediately guess how abundant a drive allele will get before being suppressed, and indeed, that answer also depends heavily on the rate of mutations to resistance. But if we reduce that cost to 1% (or even 0, values which I think are not addressed, but I did not really check), then resistance evolution is much less of an issue, and the drive allele will get very high before it goes away. (My intuition says that the drive allele never goes away if it has no fitness cost.)

So anyone broadly familiar with the process will a priori appreciate that low-cost drives get to very high frequencies in the population. And if they get to high frequencies, they will be able to invade new populations and overcome all sorts of barriers – will easily escape many hoped-for containments. I don't mean to trivialize the effort here, but if the point of this work is to propose a halt to gene drive releases on the grounds that gene drives are invasive, I'm not sure the analyses here are necessary. (In contrast, if the point was to identify realms of parameter values in which the authors thought releases were safe, then such analyses would be needed.)

One interesting outcome of this study is the demonstration that resistance will evolve quickly and suppress further spread of the drive allele even with a relatively low drive fitness cost. This result may have relevance to proposed uses of gene drives to extinguish populations.

The results are used to bolster an opinion, expressed at the end of the paper and in the Abstract (as noted above) that gene drives should not be released where there is any possibility of escape. While I might accept some justification for this opinion, it goes far beyond the work presented in the paper. Whether a drive should or should not be released depends on many social factors, including the possible good that might come from the release. It is a decision for societies, and the role of science is to inform those decisions on the possible consequences. I thus think that such an apparently bold statement puts the authors in (what I consider to be) the indefensible position of appearing so arrogant as to claim the right to impose their value judgement on the entire world – when many scientists think that gene drives could ultimately save tens of millions of lives a year. Furthermore, the paper does not actually identify any biologically serious consequence to a drive release – the drive in these models merely spreads and modifies the genome throughout the population (which some people would object to, but which may have almost no fitness consequence).

So I would suggest (= my 'opinion') that the opinion expressed be tempered accordingly and perhaps tied more closely to the findings here: "Our results suggest that drives are highly invasive under many scenarios. If there are negative consequences of drive escape, then.…". But it could certainly be interesting to watch the reaction if the paper maintains its strong, unqualified statement. If nothing else, the authors might at least label their advice as an opinion.

*Reviewer #3:*

This is a short, dense and interesting article in which the authors quantitatively predict allele frequencies in a variety of theoretical populations submitted to artificial gene drive. The article limits its investigation to sexual populations of diploid individuals and to alteration-type drives. The general conclusion is that the drives have high probabilities of being invasive in wild populations, even highly structured ones, unless the gene flow between subpopulations is very low. This is true even if the homing efficiency is low. Based on their quantitative simulations, the authors conclude that presently available drive systems should not be field-tested, contrary to recent conclusions of the National academies of Sciences, Engineering and Medicine. This issue is serious enough to merit publication of this article.

I found the article well documented on the CRISPR gene drive systems and the recently published laboratory assays. Their analysis presented in Appendix is very useful.

My only concern about this work is that all computations were made with the hypothesis that resistant mutations were neutral. This may be true for the experimental models reported but cannot be considered as universal. A fitness cost of the resistant mutations would immediately alter the results in Figure 1D and in Figure 2, for example, and I would urge the authors to take this parameter in consideration.

Incidentally, in the first natural gene-drive ever reported, the group I intron of yeast mitochondrial DNA discovered nearly 40 years ago, resistant mutations to the homing-endonuclease had a major cost because they fall into the peptidyl-transferase center of rRNA. The only choice left to yeast was between being sensitive to intron invasion or being severely unfit.

---

## [Author Response]

We have made substantial revisions to the text in response, as well as including additional simulation results to address the technical comments, and we feel that the paper is greatly improved as a result. In particular, we have tuned the message in both the Abstract and Discussion, more carefully indicating opinions as such and explaining our rationale for caution regarding unwanted spread, and we have added two new figures (right panel in Figure 7 and Figure 8).

Reviewer #1:

This is an interesting paper on the dynamics of the CRISPR gene drive system. Contrary to prevalent opinion, the authors conclude that invading gene drive systems may have a lasting impact even if they do not go to fixation. In particular, substantial "peak drive" occurs under many conditions. Based on this the authors caution against being careless with such systems, particularly when it comes to introduction of gene drive systems into wild populations.The theoretical analysis is comprehensive and appears to be sound. The conclusions regarding the danger of gene drive may be a bit too sweeping.

Thank you for your positive review of our work. Given your comments as well as the comments of the other reviewers and editor, we have qualified our conclusions and altered our tone throughout the text. We feel the manuscript is greatly improved as a result.

Specific comments:- The authors start out by pointing out the need for stochastic models of populations with finite size. However, in Figure 9 they effectively show that deterministic models make the same predictions as their stochastic models (if I understand this figure correctly). So why do we need stochastic models after all? This should be better explained, since it is part of the main rationale for the analysis presented in the paper.

Thank you for pointing out this issue. To address it, we have added discussion in the text (Introduction section), clarifying that stochastic models are necessary to calculate the likelihood of invasion upon the introduction of very small numbers of organisms even when deterministic models predict invasion. While the deterministic models agree with the stochastic mean, the full distributions— including extinction events which can prevent invasion—are not captured by the deterministic models.

- The model assumes that death is random, but births occur according to fitness. What if birth is random, and death is proportional to fitness? (There are some models in which this makes a difference…)

This is a great point, thank you for bringing it to our attention. We have additionally modeled this scenario, where death rate varies based on fitness and birth rates are uniform. The results can be found in the revised Figure 7, along with discussion in the section “Effect of varying fitness and homing efficiency” and in the paragraph 11 of the Results section. In brief, we find that the mean maximum frequencies achieved in the original model (where we assumed birth rates are proportional to fitness) are approximately the same if we assume that death rates in the new model are proportional to the inverse of fitness. Essentially, it appears that the results of our model are unchanged so long as the ratio of the birth and death rates is constant. While we don’t show this analytically, this is supported by the new numerical results in the revised Figure 7.

- It seems that in the model, invasion is always initialized with homozygotes? Can this assumption be relaxed?

Because our initial interest in determining invasiveness arose from concern over the possibility of escape from a laboratory or a field trial release, we assumed homozygote introduction for all single-population simulations (e.g., those in Figure 1). However, our model of invasiveness in connected populations (e.g., Figure 2) allows for heterozygote or homozygote introduction, which reflects the randomness of organism migration. Since heterozygotes exhibit self-propagating gene drive, an ideal drive system will behave identically if introduced in heterozygotes, while we would expect that less efficient drive systems would be less likely to invade upon the introduction of an equivalent number of heterozygotes.

- I don't really understand this statement, "Invasion is very unlikely when the drive is not initially favoured by selection." I thought the drive allele is never favoured by selection, as measured by the fitness values.

We thank the reviewer for pointing out this phrasing issue. This phrase arose from an unusual need to reconcile a terminology discrepancy between game-theoretic evolutionary dynamics, where “invasion” has a very specific ESS-based definition in the context of deterministic models, and ecology, where “invasion” more typically refers to spread beyond some appreciable frequency. What we intended to say in the quoted statement is that invasion (in the ecological sense) is very unlikely when there is no invasion (in the game-theoretic evolutionary dynamics sense) in deterministic models, hence our awkward phrasing. To address this issue, we have replaced the phrase “initially favored by selection” with the roughly equivalent “predicted to invade by deterministic models” or similar phrasing throughout the text.

- Why show mean and median in Figure 1E,F.

Thank you for bringing up this point. We have removed the means from these figure panels.

- I don't really understand Figure 2E: it shows that the probability of invasion into one local population depends on migration rate, but that probably should not depend on migration at all, because as I understand it, the figure presumably refers to invasion of the drive in the local population into which the drive is initially released.

We thank the reviewer for pointing out the lack of clarity in this figure panel. We have clarified the corresponding caption. This panel presents the probability of invading 1, 2, 3 or 4 additional subpopulations beyond the subpopulation in which the drive was initially released. That “originating” subpopulation is almost always invaded, since the release size in that population is large. (In fact, our choice of releasing 15 individuals into the initial population was made because it essentially ensures invasion there, based on the results from Figure 1.)

- Choosing populations proportional to the "square of total fitness" seems odd. Shouldn't it be the sum of squares of individual fitness values, or something like that?

This is a great point which does deserve more explanation, and we have added this in the section “Finite population model with population structure”. Our reasoning is that individual reproductions occur with probability proportional to the fitness of each parent (and thus proportional to the product of the two parents’ individual fitnesses), and since our choice of population should account for the total rate of reproduction in a population, it should be based on the total fitness of all possible mating-pairs, which is given by the square of the sum of individual fitness values. The related sentence in the text now reads: “We choose this population with probability proportional to the square of its total fitness, since this counts the rate of reproduction for every possible mating pair in the population (as matings occur with rates proportional to the fitness of each parent).”

Reviewer #2:

Conceptually, I view this paper as having two parts. The first is an analysis of gene drive models to study the spread of drives and consequent evolution of resistance to the drives. The second part is a value judgement on the deployment of drives. The latter occupies little space in the paper, but is extreme and certain to attract attention ("Contrary to the National Academies report on gene drive, our results suggest that standard drive systems should not be developed nor field-tested in regions harboring the host organism." is the last sentence of the Abstract). For convenience, I will refer to these as Part I and II, even though the paper is not structured that way.Part I is possibly the most comprehensive analysis to date of drive and resistance evolution under alternative population structure scenarios. The overall message is that drives are highly invasive, albeit only temporary, despite resistance evolution. The quantitative details are not intuitive, but I would say the qualitative conclusion is obvious. Most of the models here assume a drive fitness cost of 10%, and what is not obvious is now far a drive allele would rise before being shut down by resistance evolution – the resistance evolution is assured by the fitness cost of the drive. So one cannot immediately guess how abundant a drive allele will get before being suppressed, and indeed, that answer also depends heavily on the rate of mutations to resistance. But if we reduce that cost to 1% (or even 0, values which I think are not addressed, but I did not really check), then resistance evolution is much less of an issue, and the drive allele will get very high before it goes away. (My intuition says that the drive allele never goes away if it has no fitness cost.)

We thank Jim for his forthright and insightful review and willingness to identify himself. We address the technical concerns below particular comments here (i.e., the comments regarding Part I), and at the end of our response we address the Part II comments.

Regarding the fitness issue, it is correct that many of the results assume a 10% drive fitness cost. However, we have an additional section later into the paper (expanded in this revision) which varies the drive fitness cost as well as the cost of resistance: “Effect of varying fitness and homing efficiency”. According to our results, your intuition is exactly right: when the fitness cost of the drive is very low, then it can attain very high frequencies (often fixing) given efficient homing (Figure 7). We additionally explore a modified model where the effect of fitness cost is to increase individuals’ death rate rather than decrease their birth rate, and we find that the effect is very similar. We also find that assigning a fitness cost to resistance markedly increases the maximum frequency achieved by the drive if the cost of the drive is less than the cost of resistance (Figure 8).

Aside from these details, our primary aim was to explore qualitative behaviors, which are similar across a large range of drive fitness costs, so we didn’t vary this parameter in many of the figures, instead devoting the above-mentioned section to exploring these effects.

So anyone broadly familiar with the process will a priori appreciate that low-cost drives get to very high frequencies in the population. And if they get to high frequencies, they will be able to invade new populations and overcome all sorts of barriers – will easily escape many hoped-for containments. I don't mean to trivialize the effort here, but if the point of this work is to propose a halt to gene drive releases on the grounds that gene drives are invasive, I'm not sure the analyses here are necessary. (In contrast, if the point was to identify realms of parameter values in which the authors thought releases were safe, then such analyses would be needed.)

We certainly agree that low-cost drives can attain high frequencies and overcome all sorts of barriers, but we consider it to be non-obvious to most, particularly in light of recent discussion regarding drive resistance. In particular, we find that long-term stability (impacted by resistance) is often conflated with the potential for invasiveness or lack thereof (not significantly impacted by resistance). Also, we found it surprising how costly the drive can be while still attaining high frequencies following very small releases.

One interesting outcome of this study is the demonstration that resistance will evolve quickly and suppress further spread of the drive allele even with a relatively low drive fitness cost. This result may have relevance to proposed uses of gene drives to extinguish populations.

Thank you for this positive assessment of our work, we agree that this is surprising and that there is much additional work to be done regarding suppression drives which aim to extinguish populations.

The results are used to bolster an opinion, expressed at the end of the paper and in the Abstract (as noted above) that gene drives should not be released where there is any possibility of escape. While I might accept some justification for this opinion, it goes far beyond the work presented in the paper. Whether a drive should or should not be released depends on many social factors, including the possible good that might come from the release. It is a decision for societies, and the role of science is to inform those decisions on the possible consequences. I thus think that such an apparently bold statement puts the authors in (what I consider to be) the indefensible position of appearing so arrogant as to claim the right to impose their value judgement on the entire world – when many scientists think that gene drives could ultimately save tens of millions of lives a year. Furthermore, the paper does not actually identify any biologically serious consequence to a drive release – the drive in these models merely spreads and modifies the genome throughout the population (which some people would object to, but which may have almost no fitness consequence).So I would suggest (= my 'opinion') that the opinion expressed be tempered accordingly and perhaps tied more closely to the findings here: "Our results suggest that drives are highly invasive under many scenarios. If there are negative consequences of drive escape, then.…". But it could certainly be interesting to watch the reaction if the paper maintains its strong, unqualified statement. If nothing else, the authors might at least label their advice as an opinion.

We sincerely thank Jim for this comment, and we have tuned our language accordingly throughout the text in response. We regret that we appeared to claim a right to impose our value judgments on others—that was certainly never our intention. We hope that as we clarify our motivations below, we can help to clear this issue up.

First, we believe that scientists should hold themselves morally responsible for the consequences of their work. And since two of us have put much effort toward helping popularize CRISPR-based gene drive, we consider ourselves somewhat accountable for adverse consequences of CRISPR-based gene drive. Given this, we believe that it is our obligation to look out for and address potential causes of future mistakes to avoid the negative consequences that could result.

Given this belief, three important questions arise around the topic of unauthorized spread: (1) How large is the chance of unauthorized release? (2) How likely is there to be significant spread following an unauthorized release? (3) How likely are negative consequences following unauthorized spread? As we believe the answer to all of these questions is “reasonably high”, we felt an obligation to point out these issues while recommending that we all err on the side of caution when designing and testing gene drive systems.

In retrospect, what our paper does is address the second question while implicitly assuming answers (“reasonably high”) to the first and third questions. We regret that we did not elaborate more on these points in the initial submission—and provide qualifications where we have made assumptions—so we here summarize our opinions, and we have also revised the text to reflect these points, while being sure to label opinions as such.

Toward question 1, we recall a historical example: the illegal introduction of rabbit haemorrhagic disease into New Zealand. This was done by individuals simply for economic reasons, and we consider it highly likely that gene drive systems focused on pest suppression could be similarly tempting for individuals to transport without authorization. Thus, we consider the likelihood of unauthorized release to be underappreciated in at least some circumstances.

Toward question 3, we believe that the majority of drive systems would not produce adverse effects, per se—ecological or otherwise—following unauthorized release. However, we believe that unauthorized release could harm the future potential of the field by prompting (well-deserved) public backlash and rendering the kind of international agreement required to deploy a self-propagating drive system much more difficult. As a historical example of this point, we recall the decade-plus delay in gene therapy that resulted from the tragic death of Jesse Gelsinger.

While neither of these points are analyzed in this paper (or elsewhere), we feel that they are sufficiently self-evident to at least warrant an abundance of caution. Hence, we may have erred on the side of alarmism. Besides significant edits to the Abstract and existing discussion, we have attempted to concisely state our beliefs above by including the following paragraph in the Discussion:

“These findings raise two important questions: (1) How likely are unauthorized releases of self-propagating gene drive systems in the first place? (2) How likely are serious negative consequences given the apparently high likelihood of spread to most populations of the target species? Rigorously addressing these questions is an important direction for future work, and we can offer only opinions here. The answer to the first question likely depends on a large number of factors, such as species, application, containment strategies, economic motivations, drive development stages, geography, and the caution of the investigators, so we omit speculation here. However, we consider the answer to the second question to be clearer: although most laboratory gene drive systems are unlikely to cause ecological changes—they are typically predicted to be transient and are not designed to alter traits of the host organism, least of all interactions with other species—the history of genetic engineering offers many examples suggesting that substantial social backlash could be triggered by unauthorized spread of a self-propagating gene drive. Any such event could significantly reduce public support for interventions against diseases such as malaria that could possibly save millions of lives. We believe it would be profoundly unwise to proceed with anything less than an abundance of caution.”

In addition, to address Jim's point above and other scientists who are concerned by the possible conflation of alteration and suppression drive, we have added a paragraph to the Discussion section clarifying that our results are specific to alteration drive and briefly outlining similarities and differences with respect to the potential invasiveness of suppression drive. Other studies have shown that even extremely low levels of resistance will prevent populations from being extinguished.

Reviewer #3:

This is a short, dense and interesting article in which the authors quantitatively predict allele frequencies in a variety of theoretical populations submitted to artificial gene drive. The article limits its investigation to sexual populations of diploid individuals and to alteration-type drives. The general conclusion is that the drives have high probabilities of being invasive in wild populations, even highly structured ones, unless the gene flow between subpopulations is very low. This is true even if the homing efficiency is low. Based on their quantitative simulations, the authors conclude that presently available drive systems should not be field-tested, contrary to recent conclusions of the National academies of Sciences, Engineering and Medicine. This issue is serious enough to merit publication of this article.I found the article well documented on the CRISPR gene drive systems and the recently published laboratory assays. Their analysis presented in Appendix is very useful.

We thank Bernard for his positive and enthusiastic review of our work, as well as his willingness to reveal his identity.

My only concern about this work is that all computations were made with the hypothesis that resistant mutations were neutral. This may be true for the experimental models reported but cannot be considered as universal. A fitness cost of the resistant mutations would immediately alter the results in Figure 1D and in Figure 2, for example, and I would urge the authors to take this parameter in consideration.

Thank you for bringing this point to our attention. We initially did not vary this parameter because we felt that neutral resistance was the most typical scenario for proof-of-concept drive constructs that have been engineered in the past and also because it represents the most conservative scenario for invasiveness (i.e., one might expect containment to be easier when resistance is neutral than when it is highly deleterious). However, drive constructs constructed for applications would likely be designed to feature selection against resistant alleles, which could be expected to increase their invasiveness, so you are absolutely right that this parameter should be considered. Accordingly, we have included a new figure (Figure 8) in the section “Effect of varying fitness and homing efficiency”, where we explore the effect of varying the fitness cost of resistance as well as the fitness cost of the drive. Related discussion can be found in the paragraph 11 of the Results section as well as the section noted above. Essentially, we find that if the fitness cost of resistance is lower than that of the drive, then drive spread can be dramatically increased, particularly when both costs are low.

Incidentally, in the first natural gene-drive ever reported, the group I intron of yeast mitochondrial DNA discovered nearly 40 years ago, resistant mutations to the homing-endonuclease had a major cost because they fall into the peptidyl-transferase center of rRNA. The only choice left to yeast was between being sensitive to intron invasion or being severely unfit.

Thank you for pointing out this interesting example of costly resistance in a natural drive system! This certainly underscores the point that costly resistance is something important to consider, and we believe the paper has been improved with the addition of this new analysis. Thank you again for bringing this to our attention.